# ZeroDiff: Zero-Shot Time Series Reconstruction via Informed-Prior Diffusion

**Yingda Fan** [1]  **Dan Lu** [2]  **Xiaowei Jia** [1]

## Abstract

Time series modeling increasingly demands high-quality supervision, yet target observations remain scarce—exogenous inputs are broadly available, but target measurements are often unavailable due to cost, infrastructure, or accessibility constraints. Can models trained on observed locations reconstruct target time series where measurements have never been collected? We term this *zero-shot time series reconstruction*. A naive approach—directly mapping exogenous inputs to targets—can yield predictions at unobserved locations, but without target signals, such models fail to capture the intrinsic dynamics of the target variable, producing overly smooth outputs that underestimate extremes. This reveals systematic errors that call for explicit modeling and calibration. We propose **ZeroDiff**, which constructs an informed prior from exogenous variables alone, then learns to calibrate reconstruction errors through diffusion—training on observed locations and generalizing to unobserved ones. Experiments across diverse real-world datasets demonstrate significant improvements over existing approaches. Our code is available at https://github.com/YingdaFan/ZeroDiff-ICML2026.

## 1. Introduction

As time series modeling scales from specialized architectures (Nie et al., 2023; Liu et al., 2024) to foundation models (Ansari et al., 2024; Das et al., 2024), the need for high-quality supervision has grown—yet in many scenarios, target observations remain scarce or entirely missing. This challenge arises across domains—from flood forecasting (Kratzert et al., 2018), where meteorological data are abundant but streamflow is sparsely observed across space, to solar energy prediction (Paletta et al., 2023), where satel-

lite imagery and reanalysis data are globally available but irradiance observations are measured at few stations. In each case, exogenous inputs are broadly available, but target measurements are observed only at a subset of locations. In such settings, practitioners face a fundamental question: can models learn from locations where targets are observed to reconstruct target time series at locations where they have never been measured? We term this problem *zero-shot cross-domain time series reconstruction*. Unlike forecasting (Zhou et al., 2021; Salinas et al., 2020), which predicts future values from historical observations at the same location, or imputation (Cao et al., 2018; Du et al., 2023), which fills in missing values given partial observations, zero-shot reconstruction requires the model to infer an entire target time series at locations with no historical target data whatsoever. This setting poses a distinct challenge: the model must generalize across locations, reconstructing temporal dynamics it has never directly observed.

A central difficulty lies in the heterogeneity of target distributions across locations. While the temporal dynamics—how targets respond to exogenous inputs—are often governed by shared physical or statistical principles, the marginal distribution of the target variable can vary dramatically in mean and variance. As a result, models trained on observed locations struggle to transfer to unobserved ones.

In such settings, probabilistic reconstruction is essential. When ground truth targets are unavailable, uncertainty quantification becomes important for downstream decision-making (Gneiting & Katzfuss, 2014). Diffusion models (Ho et al., 2020; Song et al., 2021) offer a principled framework for probabilistic modeling (Rasul et al., 2021; Tashiro et al., 2021), but standard diffusion requires access to $\mathbf{Y}_0$ for both forward corruption and loss computation—impossible when targets are absent. Recent work shows diffusion benefits from informative initialization: CARD (Han et al., 2022) replaces $\mathcal{N}(\mathbf{0}, \mathbf{I})$ with a regression-based prior, SDEdit (Meng et al., 2022) preserves structure via partial noising. These advances suggest that if we can construct an informed prior $\hat{\mathbf{Y}}$ capturing the expected target, diffusion can shift from generation to calibration—refining a structured estimate rather than generating from noise.

We present **ZeroDiff**, a diffusion-based framework for zero-shot time series reconstruction. It decomposes the cross-location generalization problem into two stages connected

---

[1]Department of Computer Science, University of Pittsburgh, Pittsburgh, PA, USA [2]Oak Ridge National Laboratory, Oak Ridge, TN, USA. Correspondence to: Xiaowei Jia <xiaowei@pitt.edu>.

*Proceedings of the 43rd International Conference on Machine Learning*, Seoul, South Korea. PMLR 306, 2026. Copyright 2026 by the author(s).

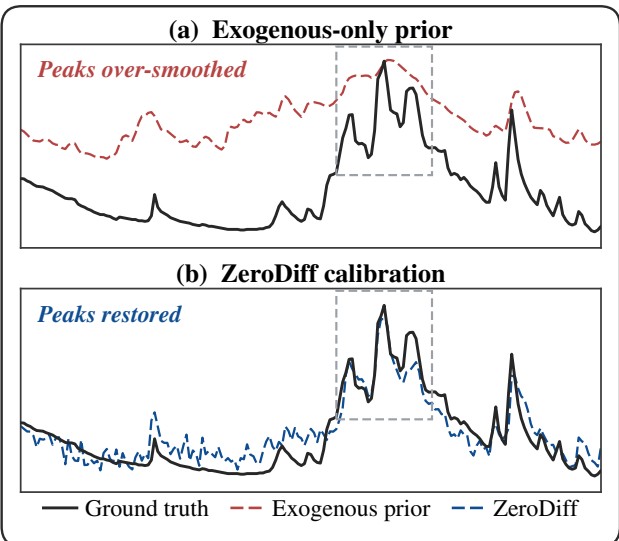

*Figure 1.* In the exogenous-only reconstruction the peaks are over-smoothed and no longer stand out **(a)**; ZeroDiff calibrates it onto the ground truth, restoring their prominence **(b)**.

by a shared informed prior $\hat{\mathbf{Y}}$. The first stage generalizes from exogenous inputs to target structure: target distributions vary drastically across locations in mean and variance, but their input–output dynamics are shared once expressed in a standardized space. The location-specific statistics needed for that standardization can themselves be inferred from exogenous variables alone. The informed prior $\hat{\mathbf{Y}}$ at unobserved locations is then constructed directly from $\mathbf{X}$, without any target signal.

The second stage generalizes calibration across locations. Because $\hat{\mathbf{Y}}$ is constructed identically at observed and unobserved locations, its systematic errors share the same structure everywhere. A diffusion model trained to calibrate the prior on observed locations therefore transfers to unobserved ones. This shifts the role of diffusion from generation to calibration: the reverse process refines $\hat{\mathbf{Y}}$ rather than denoising from $\mathcal{N}(\mathbf{0}, \mathbf{I})$.

In summary, our contributions are:

- We formalize the problem of **zero-shot time series reconstruction** and identify a structural assumption that makes it tractable: target magnitudes vary across locations, but input–target dynamics are shared.

- We propose **informed-prior diffusion**: a single prior inferred from exogenous variables both carries cross-location dynamics and anchors a calibration that transfers from observed to unobserved locations.

- We design a bidirectional denoiser that conditions on both past and future timesteps, improving reconstruction at peaks and sharp transitions.

**Conflict of Interest Disclosure.** The authors declare no financial conflicts of interest related to this work.

## 2. Related Work

**Diffusion Models for Time Series.** Diffusion models have recently emerged as a powerful framework for probabilistic time series modeling, achieving strong performance across forecasting, imputation, and generation tasks. TimeGrad (Rasul et al., 2021) first introduces autoregressive denoising diffusion for probabilistic forecasting, generating future values step-by-step conditioned on historical observations. CSDI (Tashiro et al., 2021) extends score-based diffusion to imputation via conditional generation. SSSD (Alcaraz & Strodthoff, 2022) integrates structured state-space models to capture long-range dependencies. TimeDiff (Shen & Kwok, 2023) proposes non-autoregressive diffusion for parallel generation. Diffusion-TS (Yuan & Qiao, 2024) employs encoder-decoder transformers with disentangled temporal representations. mr-Diff (Shen et al., 2024b) leverages multi-resolution structure through seasonal-trend decomposition. Most recently, CNDiff (Rishi et al., 2025) introduces nonlinear data transformations for conditional forecasting. Despite these advances, all existing diffusion-based time series methods share a fundamental assumption: target observations must be available at locations during training. This assumption is embedded at multiple levels—normalization, forward corruption, and loss computation—all of which require access to ground-truth targets. Consequently, when targets are entirely absent at a subset of locations, the diffusion framework cannot be applied. Our work addresses this previously unsupported regime.

**Diffusion with Informative Priors.** Standard diffusion models assume the forward process terminates at an uninformative prior $\mathcal{N}(\mathbf{0}, \mathbf{I})$ (Ho et al., 2020; Song et al., 2021), requiring the reverse process to generate from scratch. Recent work in computer vision has challenged this assumption by incorporating *data-dependent priors*. PriorGrad (Lee et al., 2022) demonstrates that adaptive priors derived from conditional information improve both efficiency and sample quality. CARD (Han et al., 2022) replaces the standard end-point with a regression-based prior $\mathcal{N}(f(\mathbf{x}), \sigma)$, where $f(\mathbf{x})$ is a pre-trained predictor; the diffusion process then *refines* predictions rather than generates from noise. SDEdit (Meng et al., 2022) starts the reverse process from partially noised inputs, preserving structural information from a guide image. Cold Diffusion (Bansal et al., 2023) demonstrates that meaningful degradations—such as blur or downsampling—can replace Gaussian noise entirely, revealing that diffusion's core mechanism is learning to invert transformations. These approaches share a common insight: diffusion is most effective as a refinement mechanism when initialized from a structured prior, a design principle that has also

influenced other prior-guided reconstruction pipelines (Yuan et al., 2025b;a). However, existing methods assume that the prior itself is obtained from supervised learning using target observations at the same domain. In contrast, our setting requires constructing informative priors without any target data at the reconstruction location. **ZeroDiff** extends the idea of informative priors to a zero-shot regime, where priors are inferred solely from exogenous variables and calibrated via diffusion using supervision from other locations.

**Zero-shot Learning and Cross-location Generalization.**
Zero-shot learning enables recognition of classes unseen during training by leveraging auxiliary information (Lampert et al., 2009). Early approaches learn mappings between visual features and semantic attributes (Farhadi et al., 2009) or word embeddings (Frome et al., 2013). Subsequent work explores compatibility functions (Akata et al., 2016), latent embeddings (Xian et al., 2016), and generative approaches (Xian et al., 2018). Xian et al. (Xian et al., 2017) provide a comprehensive benchmark establishing evaluation protocols for the field. The core principle—transferring knowledge from seen to unseen classes via shared semantic representations—motivates our approach of transferring dynamics from observed to unobserved locations.

In spatio-temporal modeling, cross-location generalization has been explored through graph neural networks. STGCN (Yu et al., 2018) and DCRNN (Li et al., 2018) capture spatial dependencies via graph convolutions. Recent work addresses cross-city transfer: Domain Adversarial Spatial-Temporal Networks (Tang et al., 2022) learn city-invariant representations, while Graph Neural Processes (Hu et al., 2023) enable extrapolation to unobserved locations. However, these methods assume that at least some target observations are available at all locations, focusing on forecasting rather than reconstruction. They do not address the setting where targets are unobserved at test locations.

Time series foundation models such as Chronos (Ansari et al., 2024), TimesFM (Das et al., 2024), and Time-MoE (Shi et al., 2025) demonstrate impressive zero-shot capabilities across datasets. However, their "zero-shot" refers to generalization across *datasets*—not across locations within a dataset where some locations lack target observations entirely.

In hydrology, the problem of Prediction in Ungauged Basins (PUB) (Kratzert et al., 2019) is conceptually related, aiming to predict streamflow at locations without gauges. While recent machine learning approaches improve point estimates in this setting, they typically lack principled uncertainty quantification.

# 3. Preliminary

## 3.1. Problem Formulation

**Zero-shot time series reconstruction.** Given exogenous variables $\mathbf{X}$ at a new location, the goal is to reconstruct the target time series $\mathbf{Y}$ without any historical target observations at that location.

Formally, consider $K$ locations, each with exogenous variables $\mathbf{X}^{(k)} \in \mathbb{R}^{L \times D}$ and target time series $\mathbf{Y}^{(k)} \in \mathbb{R}^L$, where $\mathbf{X}$ and $\mathbf{Y}$ are temporally aligned with sequence length $L$ (Tashiro et al., 2021; Cao et al., 2018). Locations are partitioned into:

- **Observed** $\mathcal{O}$: both $\mathbf{X}^{(k)}$ and $\mathbf{Y}^{(k)}$ are available for training.

- **Unobserved** $\mathcal{U}$: only $\mathbf{X}^{(j)}$ is accessible; the goal is to reconstruct $\mathbf{Y}^{(j)}$.

Both training and inference operate on the same temporal span. This differs from forecasting (Zhou et al., 2021; Wu et al., 2021; Nie et al., 2023; Liu et al., 2024), which predicts future from past at the *same* location, and from imputation (Tashiro et al., 2021; Cao et al., 2018; Du et al., 2023), which fills gaps given *partial* observations. Here, the model must generalize across locations to reconstruct targets it has never seen.

The central challenge is that while the input-target dynamics may be shared across locations once properly standardized, the marginal distribution of $\mathbf{Y}$ differs substantially in mean and variance across locations. Standard normalization techniques (Kim et al., 2022; Liu et al., 2022) require target observations to compute location-specific statistics $(\mu, \sigma)$—precisely what is unavailable for $j \in \mathcal{U}$.

## 3.2. Diffusion Models for Time Series

Denoising diffusion probabilistic models (DDPMs) (Ho et al., 2020; Song et al., 2021; Luo, 2022) learn data distributions through a forward process that progressively adds noise and a reverse process that learns to denoise. Given target $\mathbf{Y}_0$, the forward process is defined as:

$$q(\mathbf{Y}_t|\mathbf{Y}_0) = \mathcal{N}\big(\sqrt{\bar{\alpha}_t}\mathbf{Y}_0, (1 - \bar{\alpha}_t)\mathbf{I}\big) \qquad (1)$$

The reverse process learns to denoise by optimizing $\mathcal{L}_{\text{DDPM}} = \mathbb{E}_{t,\mathbf{Y}_0,\epsilon}\big[\|\epsilon - \epsilon_\theta(\mathbf{Y}_t, t)\|^2\big]$. For time series applications, two formulations are common depending on whether temporal windows are employed:

**Sliding window formulation.** In forecasting tasks (Rasul et al., 2021; Li et al., 2024; Shen et al., 2024a; Yuan & Qiao, 2024; Gao et al., 2025), the model conditions on historical observations $\mathbf{X} \in \mathbb{R}^{N \times D}$ to predict future targets

$\mathbf{Y} \in \mathbb{R}^{M \times D}$, where $N$ and $M$ denote the lookback and prediction horizon lengths respectively.

**Aligned formulation.** In reconstruction or imputation tasks (Tashiro et al., 2021; Alcaraz & Strodthoff, 2023; Kollovieh et al., 2023), $\mathbf{X} \in \mathbb{R}^{L \times D}$ and $\mathbf{Y} \in \mathbb{R}^{L}$ share the same temporal span $L$. The model learns to recover $\mathbf{Y}$ from temporally aligned exogenous inputs.

A key advancement is the use of *informed priors* (Han et al., 2022; Meng et al., 2022; Li et al., 2024). Rather than diffusing toward an uninformative endpoint $\mathcal{N}(\mathbf{0}, \mathbf{I})$, these methods use a regression-based prior $\mathcal{N}(\hat{\mathbf{Y}}, \sigma^2\mathbf{I})$, where $\hat{\mathbf{Y}} = f(\mathbf{X})$. The forward process becomes:

$$q(\mathbf{Y}_t|\mathbf{Y}_0, \mathbf{X}) = \mathcal{N}\left(\sqrt{\bar{\alpha}_t}\mathbf{Y}_0 + (1 - \sqrt{\bar{\alpha}_t})\hat{\mathbf{Y}}, \bar{\sigma}_t\mathbf{I}\right) \quad (2)$$

where $\bar{\sigma}_t = 1 - \bar{\alpha}_t$.

**The missing-target problem.** Crucially, both formulations require access to ground-truth targets $\mathbf{Y}_0$ during training. The forward process in Eq. (2) constructs $\mathbf{Y}_t$ from $\mathbf{Y}_0$, and the denoising process computes the loss using $\mathbf{Y}_t$. For unobserved locations, neither the forward process nor the training objective can be defined—the diffusion framework fundamentally breaks down.

## 4. Methodology

We propose a framework for zero-shot time series reconstruction, illustrated in Fig. 2, built on three insights: (i) while target marginals $p(\mathbf{Y})$ vary drastically across locations in mean and standard deviation, the input-target dynamics are transferable once standardized; (ii) location-specific statistics $(\mu, \sigma)$ can be inferred from exogenous variables $\mathbf{X}$ alone via cross-modal learning, without requiring target observations; (iii) by constructing informed priors for $\mathcal{O}$ where ground-truth is available, the diffusion model learns to calibrate estimation errors on $\mathcal{O}$, and this calibration generalizes to $\mathcal{U}$. Motivated by these observations, we first construct an informed prior through moment estimation and dynamics learning, and then refine it through diffusion-based calibration.

### 4.1. Transferable Prior Construction

Recent work on non-stationary time series (Kim et al., 2022; Liu et al., 2022) demonstrates that separating statistics (mean, variance) from temporal dynamics improves generalization. While the marginal distribution of $\mathbf{Y}$ varies across locations in magnitude, the temporal dynamics of how $\mathbf{X}$ drives $\mathbf{Y}$, once normalized, are shared. This motivates learning in normalized space, but standard normalization requires target observations to compute location-specific $(\mu, \sigma)$, which is precisely unavailable for $j \in \mathcal{U}$.

We address this limitation through *cross-modal moment es-*

*timation*: inferring target statistics $(\mu_k, \sigma_k)$ from exogenous variables $\mathbf{X}^{(k)}$ alone. Although $\mathbf{X}$ and $\mathbf{Y}$ represent different physical quantities, the distributional characteristics of $\mathbf{X}$ are predictive of the magnitude of $\mathbf{Y}$. In hydrological systems, for instance, a basin's weather data and catchment properties can affect the magnitude of runoff.

Specifically, we model the marginal moments of $\mathbf{Y}^{(k)}$ as a learnable function of exogenous summary statistics:

$$(\mu_k, \sigma_k) = h(\mathbf{Z}^{(k)}) \quad (3)$$

where $\mathbf{Z}^{(k)} = [\bar{\mathbf{X}}^{(k)}; \sigma_\mathbf{X}^{(k)}] \in \mathbb{R}^{2D}$ concatenates the temporal mean and standard deviation of exogenous inputs. Since $(\mu_k, \sigma_k)$ are time-invariant statistics, we condition on summary statistics of $\mathbf{X}$ rather than the full sequence.

**Cross-modal moment estimation.** We estimate the target moments using a conditional VAE (Kingma & Welling, 2014; Sohn et al., 2015)—a form of exogenous-conditioned prior estimation that captures distributional structure across locations. The architecture consists of three components: a *latent encoder* $q_\psi(\mathbf{z}|\mu_k, \sigma_k)$ that maps target moments to a latent distribution, capturing location-specific deviations from the mean behavior; a *feature encoder* $\phi(\mathbf{Z}^{(k)})$ that transforms exogenous statistics into a representation conditioning the decoder; and a *decoder* $p_\psi(\mu_k, \sigma_k|\mathbf{z}, \phi(\mathbf{Z}^{(k)}))$ that reconstructs target moments from both the latent code and the encoded exogenous features.

During training on $\mathcal{O}$, we optimize the evidence lower bound:

$$\begin{aligned}
\mathcal{L}_{\text{VAE}} = \mathbb{E}_{q_\psi} & \left[\|(\mu_k, \sigma_k) - \text{Dec}_\psi(\mathbf{z}, \phi(\mathbf{Z}^{(k)}))\|^2\right] \\
& + \lambda \cdot D_{\text{KL}}\left(q_\psi(\mathbf{z}|\mu_k, \sigma_k) \| \mathcal{N}(\mathbf{0}, \mathbf{I})\right)
\end{aligned} \quad (4)$$

The KL regularization encourages the latent space to capture shared distributional structure: locations with similar exogenous characteristics are mapped to nearby points, enabling moment estimation to generalize from $\mathcal{O}$ to $\mathcal{U}$.

For $j \in \mathcal{U}$, although $\mathbf{Y}^{(j)}$ is unavailable, we can compute $\mathbf{Z}^{(j)} = [\bar{\mathbf{X}}^{(j)}; \text{std}(\mathbf{X}^{(j)})]$ directly from $\mathbf{X}^{(j)}$. We then decode using the prior mean:

$$(\hat{\mu}_j, \hat{\sigma}_j) = \text{Dec}_\psi(\mathbf{0}, \phi(\mathbf{Z}^{(j)})) \quad (5)$$

This enables zero-shot moment estimation: using only $\mathbf{X}^{(j)}$ and the distributional structure learned from $\mathcal{O}$, we infer $(\hat{\mu}_j, \hat{\sigma}_j)$ without any target observations.

**Dynamics learning in standardized space** We learn a *single* dynamics model $f_\omega : \mathbb{R}^{L \times D} \to \mathbb{R}^L$ shared across all observed locations, performing reconstruction over the same temporal span rather than forecasting. The dynamics model $f_\omega$ can adopt architectures such as LSTM (Kratzert

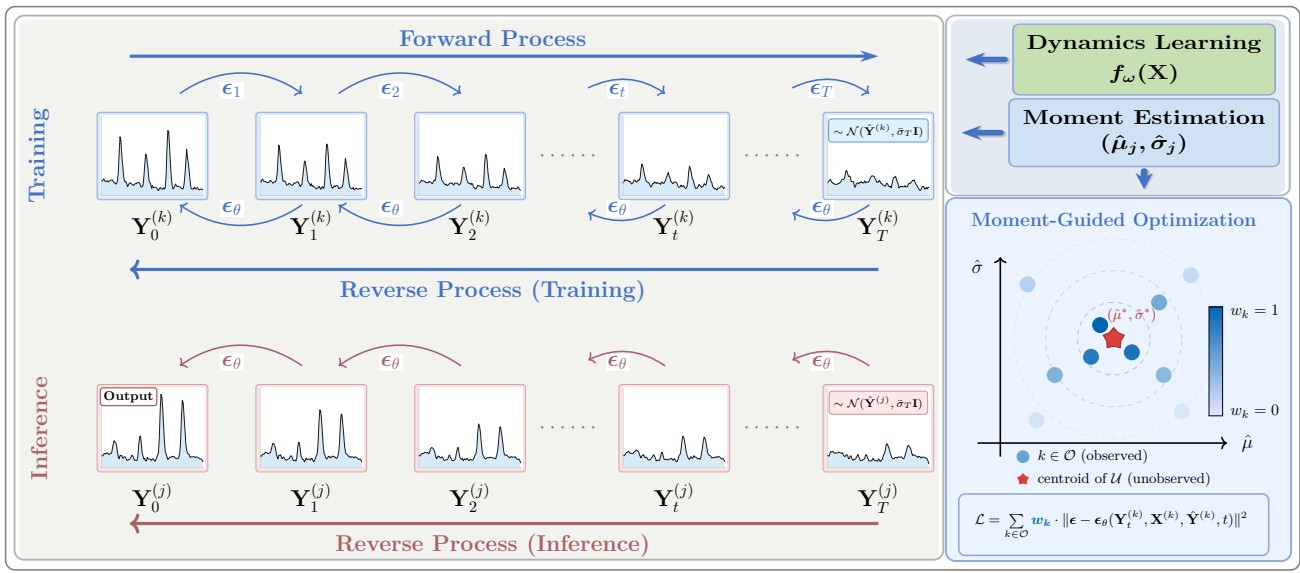

*Figure 2.* Proposed framework. The informed prior $\hat{\mathbf{Y}} = \Phi_{\psi,\omega}(\mathbf{X})$ combines moment estimation (VAE) with dynamics learning in standardized space, generalizing to unobserved locations. The diffusion process starts from $\mathcal{N}(\hat{\mathbf{Y}}, \bar{\sigma}_T\mathbf{I})$ rather than pure noise, enabling calibration instead of generation. The estimated moments also guide optimization by weighting training locations based on their proximity to the target in moment space.

et al., 2018), DLinear (Zeng et al., 2023), Mamba (Gu & Dao, 2023), or TimesNet (Wu et al., 2023); we compare these choices in Table 2.

The training objective aggregates prediction error over $\mathcal{O}$:

$$\min_{\omega} \sum_{k \in \mathcal{O}} \left\| f_{\omega}(\mathbf{X}^{(k)}) - \tilde{\mathbf{Y}}^{(k)} \right\|^2 \qquad (6)$$

where $\tilde{\mathbf{Y}}^{(k)} = (\mathbf{Y}^{(k)} - \mu_k)/\sigma_k$ is the standardized target. By sharing parameters and training in normalized space, $f_{\omega}$ captures the shared temporal dynamics of how $\mathbf{X}$ drives fluctuations in $\mathbf{Y}$, rather than fitting location-specific magnitudes.

**Informed prior construction.** We use VAE-estimated statistics $(\hat{\mu}, \hat{\sigma})$ for both $\mathcal{O}$ and $\mathcal{U}$—even for $k \in \mathcal{O}$ where ground-truth statistics $(\mu_k, \sigma_k)$ are available. This ensures consistent estimation of error characteristics between training and inference, enabling the downstream diffusion model to learn a calibration that generalizes to $\mathcal{U}$. The informed prior for $k \in \mathcal{O}$:

$$\hat{\mathbf{Y}}^{(k)} = \hat{\mu}_k + \hat{\sigma}_k \cdot f_{\omega}(\mathbf{X}^{(k)}) \qquad (7)$$

For $j \in \mathcal{U}$:

$$\hat{\mathbf{Y}}^{(j)} = \hat{\mu}_j + \hat{\sigma}_j \cdot f_{\omega}(\mathbf{X}^{(j)}) \qquad (8)$$

where $(\hat{\mu}_j, \hat{\sigma}_j)$ is obtained via Eq. (5). For unobserved locations, the estimated moments play a critical role: $f_{\omega}$ captures the normalized response pattern, while $(\hat{\mu}_j, \hat{\sigma}_j)$ rescales it to the appropriate magnitude. For notational convenience, we denote the composite mapping as $\Phi_{\psi,\omega}(\mathbf{X}) := \hat{\mu} + \hat{\sigma} \cdot f_{\omega}(\mathbf{X})$, where $(\hat{\mu}, \hat{\sigma}) = \mathrm{Dec}_{\psi}(\mathbf{0}, \phi(\mathbf{Z}))$.

### 4.2. Generalizable Diffusion Calibration

The informed prior $\hat{\mathbf{Y}}^{(k)}$ and $\hat{\mathbf{Y}}^{(j)}$ provide an initial reconstruction but inherit estimation error from both moment estimation and dynamics learning. We refine this prior through diffusion, learning the conditional distribution $p(\mathbf{Y}^{(k)}|\hat{\mathbf{Y}}^{(k)}, \mathbf{X}^{(k)})$ on $\mathcal{O}$. The denoiser $\epsilon_{\theta}$ is shared across all $k \in \mathcal{O}$, learning a calibration that generalizes to $\mathcal{U}$. At inference, we apply this learned distribution to $j \in \mathcal{U}$ by conditioning on $(\hat{\mathbf{Y}}^{(j)}, \mathbf{X}^{(j)})$. Since the estimation procedure is shared, the error characteristics remain consistent across $\mathcal{O}$ and $\mathcal{U}$, enabling the calibration to generalize.

**Informed-prior forward process.** We diffuse toward the informed prior rather than zero:

$$q(\mathbf{Y}_t^{(k)}|\mathbf{Y}^{(k)}, \mathbf{X}^{(k)}) = \mathcal{N}\big(\sqrt{\bar{\alpha}_t}\mathbf{Y}^{(k)} + (1-\sqrt{\bar{\alpha}_t})\hat{\mathbf{Y}}^{(k)}, \bar{\sigma}_t\mathbf{I}\big) \qquad (9)$$

The noise schedule $(\bar{\alpha}_t, \bar{\sigma}_t)$ is shared across all locations, while each location $k$ has its own informed prior $\hat{\mathbf{Y}}^{(k)} = \hat{\mu}_k + \hat{\sigma}_k \cdot f_{\omega}(\mathbf{X}^{(k)})$ as defined in Eq. (7), incorporating $\mathbf{X}^{(k)}$ through both the VAE-estimated moments and the dynamics model. At $t = T$, this yields the endpoint distribution $\mathbf{Y}_T^{(k)} \sim \mathcal{N}(\hat{\mathbf{Y}}^{(k)}, \bar{\sigma}_T\mathbf{I})$, in contrast to the standard $\mathcal{N}(\mathbf{0}, \mathbf{I})$ used in conventional diffusion models. This formulation follows the informed-prior framework of Han et al. (2022).

**Prior-anchored reverse process.** The informed-prior forward process (Eq. (9)) induces a modified posterior distribution. At each reverse step, we sample from $p_{\theta}(\mathbf{Y}_{t-1}^{(k)}|\mathbf{Y}_t^{(k)}, \mathbf{X}^{(k)}, \hat{\mathbf{Y}}^{(k)})$, whose posterior mean takes

the form:

$$\boldsymbol{\mu}_{t-1}^{(k)} = \gamma_0 \mathbf{Y}^{(k)} + \gamma_1 \mathbf{Y}_t^{(k)} + \gamma_2 \hat{\mathbf{Y}}^{(k)} \quad (10)$$

where $\gamma_0, \gamma_1, \gamma_2$ are functions of $\alpha_t, \bar{\alpha}_{t-1}$, and $\bar{\sigma}_t$ (see Appendix A). Unlike standard DDPM, our posterior includes a third term—anchoring the trajectory to $\hat{\mathbf{Y}}$.

During inference for $j \in \mathcal{U}$, the clean target $\mathbf{Y}^{(j)}$ is unavailable. We estimate it via reparameterization:

$$\hat{\mathbf{Y}}_0^{(j)} = \frac{1}{\sqrt{\bar{\alpha}_t}} \big( \mathbf{Y}_t^{(j)} - (1 - \sqrt{\bar{\alpha}_t}) \hat{\mathbf{Y}}^{(j)} - \sqrt{\bar{\sigma}_t} \boldsymbol{\epsilon}_\theta \big) \quad (11)$$

Substituting $\hat{\mathbf{Y}}_0^{(j)}$ for $\mathbf{Y}^{(j)}$ in Eq. (10) yields $\boldsymbol{\mu}_{t-1}^{(j)}$, ensuring the reverse trajectory remains guided by the informed prior throughout denoising.

**Moment-guided optimization.** In addition to denormalization in Eq. (8), the estimated moments $(\hat{\mu}_k, \hat{\sigma}_k)$ guide the optimization of the shared denoiser $\boldsymbol{\epsilon}_\theta$. Since every $k \in \mathcal{O}$ contributes to updating the shared denoiser parameters, we reweight each location's gradient contribution based on its proximity to the target in moment space. Let $(\hat{\mu}^*, \hat{\sigma}^*)$ denote the estimated moments of the target location; for simultaneous reconstruction of multiple targets, we use the centroid (Appendix E.1). We define weights using a Gaussian kernel over the moment space:

$$w_k = \exp \left( -\frac{\|(\hat{\mu}_k, \hat{\sigma}_k) - (\hat{\mu}^*, \hat{\sigma}^*)\|^2}{2\tau^2} \right) \quad (12)$$

where $\tau$ controls the bandwidth. The training objective becomes:

$$\mathcal{L} = \sum_{k \in \mathcal{O}} w_k \cdot \mathbb{E}_{t,\boldsymbol{\epsilon}} \left[ \|\boldsymbol{\epsilon} - \boldsymbol{\epsilon}_\theta(\mathbf{Y}_t^{(k)}, \mathbf{X}^{(k)}, \hat{\mathbf{Y}}^{(k)}, t)\|^2 \right] \quad (13)$$

Rather than aggregating gradients equally across all $k \in \mathcal{O}$, this formulation upweights locations whose moments are close to $(\hat{\mu}^*, \hat{\sigma}^*)$, yielding a denoiser better suited for $j \in \mathcal{U}$.

**Bidirectional denoiser.** Our calibration setting is fundamentally different from prior diffusion-based forecasting methods (Rasul et al., 2021), which adopts causal masking to prevent information leakage from future to past (Tashiro et al., 2021; Shen & Kwok, 2023). In contrast, during the calibration process $p(\mathbf{Y}^{(k)}|\hat{\mathbf{Y}}^{(k)}, \mathbf{X}^{(k)})$, the conditioning signals $\mathbf{X}^{(k)}$ and $\hat{\mathbf{Y}}^{(k)}$ are fully observed across all timesteps. This enables *bidirectional conditioning*—realized through full self-attention, information flows forward from past and backward from future simultaneously:

$$\boldsymbol{\epsilon}_\theta(\mathbf{Y}_{t,\ell}^{(k)}) = f_\theta\big(\mathbf{Y}_{t,1:L}^{(k)}, \mathbf{X}_{1:L}^{(k)}, \hat{\mathbf{Y}}_{1:L}^{(k)}, t\big), \quad \forall \ell \in \{1, \dots, L\} \quad (14)$$

This bidirectional flow facilitates the reconstruction of complex temporal dynamics that are difficult to infer from either

direction alone. For example, a peak can be identified when the forward view captures an upward trend before a certain timestep and the backward view captures a downward trend immediately after it. The effect of the bidirectional flow is validated by the ablation results in Table 1.

# 5. Experiments

## 5.1. Experimental Setup

*Table 3.* Dataset statistics. Each experiment masks one location and trains on the rest.

|  | STREAMFLOW | SOLAR | TEMP | METHANE |
|---|---|---|---|---|
| Domain | Hydrology | Energy | Hydrology | Carbon Emission |
| Exog. Dim | 33 | 48 | 8 | 16 |
| Length | 4745 | 4015 | 1095 | 4015 |
| Test Loc. | 100 | 30 | 42 | 30 |

**Datasets.** We evaluate on four datasets spanning four scientific domains (Table 3). Unlike standard forecasting benchmarks that split data temporally, zero-shot reconstruction requires *spatial* partitioning: we evaluate whether models trained on observed locations can generalize to held-out ones. For each dataset, we adopt a leave-one-location-out protocol—iteratively masking all target observations at one location while retaining its exogenous inputs, training on the remaining locations, and evaluating reconstruction at the held-out location. TEST LOC. indicates how many independent trials are conducted per dataset; final metrics are averaged across all held-out locations.

CAMELS (Newman et al., 2015; Addor et al., 2017) is a widely-used hydrology benchmark containing daily streamflow from 531 gauges across the contiguous US, with catchment attributes and meteorology for large-sample studies. NHD contains daily stream water temperature derived from the National Hydrography Dataset (U.S. Geological Survey, 2019; 2024) at approximately 1-km resolution. Solar (McGovern et al., 2015) is from the AMS 2013–2014 Solar Energy Prediction Contest, with daily solar energy from Oklahoma Mesonet stations and GEFS ensemble forecasts. Methane (Pastorello et al., 2020; Sun et al., 2025) provides daily methane ($CH_4$) flux from FLUXNET.

**Fair Experiment** Most baselines are designed for time series forecasting with look-back window $L_b$ and prediction horizon $H$. To fairly evaluate on reconstruction ($L = 365$), we adopt two protocols. First, sliding window reconstruction: We partition the sequence into overlapping segments. For each segment, baselines can use ground-truth $\mathbf{Y}$ from observed locations as look-back and predict the subsequent horizon. Specifically, we use $L_b \in \{96, 192, 365\}$ and $H \in \{96, 192, 365\}$, sliding with stride $H$ to cover the

*Table 1.* Performance comparison across different datasets. Best results are in **bold**. ↑ indicates higher is better, ↓ indicates lower is better.

| Model | Streamflow | | | Solar | | | Temp | | | Methane | | |
|---|---|---|---|---|---|---|---|---|---|---|---|---|
| | NSE↑ | RMSE↓ | MAE↓ | NSE↑ | RMSE↓ | MAE↓ | NSE↑ | RMSE↓ | MAE↓ | NSE↑ | RMSE↓ | MAE↓ |
| *Diffusion Baselines* | | | | | | | | | | | | |
| CSDI (Tashiro et al., 2021) | † | 2.814 | 1.267 | 0.019 | 7.896 | 6.035 | † | 6.883 | 4.525 | † | 1.910 | 1.610 |
| + $f_\omega$ | 0.060 | 2.367 | 1.214 | 0.307 | 7.271 | 5.926 | 0.722 | 2.881 | 2.286 | † | 1.229 | 1.083 |
| SSSD (Alcaraz & Strodthoff, 2022) | † | 2.583 | 1.161 | 0.446 | 5.930 | 4.468 | 0.605 | 4.002 | 3.303 | † | 0.974 | 0.890 |
| + $f_\omega$ | 0.071 | 2.348 | 1.201 | 0.374 | 6.534 | 4.865 | 0.746 | 2.756 | 2.113 | † | 1.083 | 0.986 |
| CSBI (Chen et al., 2023) | † | 2.730 | 1.446 | 0.449 | 5.915 | 4.612 | 0.243 | 6.100 | 5.260 | † | 0.976 | 0.900 |
| + $f_\omega$ | 0.084 | 2.374 | 1.270 | 0.402 | 6.410 | 4.999 | 0.725 | 2.941 | 2.439 | † | 1.083 | 0.991 |
| DiffusionTS (Yuan & Qiao, 2024) | † | 3.230 | 1.670 | 0.026 | 7.852 | 5.980 | 0.293 | 5.671 | 4.461 | † | 1.661 | 1.399 |
| + $f_\omega$ | 0.033 | 2.446 | 1.269 | 0.259 | 7.229 | 5.965 | 0.747 | 2.920 | 2.213 | † | 1.222 | 1.068 |
| NsDiff (Ye et al., 2025) | † | 3.907 | 2.686 | 0.515 | 5.546 | 4.556 | 0.700 | 2.902 | 2.371 | † | 1.597 | 1.372 |
| + $f_\omega$ | 0.105 | 2.488 | 1.325 | 0.473 | 6.131 | 4.987 | 0.763 | 2.562 | 2.063 | † | 1.262 | 1.085 |
| *Ours* | | | | | | | | | | | | |
| $f_\omega$ (Kratzert et al., 2018) | 0.131 | 2.139 | 1.099 | 0.345 | 6.407 | 5.280 | 0.813 | 2.306 | 1.848 | † | 1.074 | 0.940 |
| $\Phi_{\psi,\omega}$ | 0.481 | 1.679 | **0.725** | 0.486 | 5.707 | 4.565 | 0.852 | 2.205 | 1.752 | 0.481 | 0.371 | 0.280 |
| w/o prior | † | 2.415 | 1.440 | 0.323 | 6.470 | 5.355 | 0.829 | 2.371 | 1.902 | † | 0.848 | 0.728 |
| w/o bidir. & $w_k$ | 0.365 | 1.750 | 1.022 | 0.518 | 5.526 | 4.638 | 0.860 | 2.014 | 1.608 | 0.671 | 0.276 | 0.202 |
| w/o bidir. | 0.463 | 1.576 | 0.903 | 0.519 | 5.519 | 4.636 | 0.859 | 2.010 | 1.600 | 0.623 | 0.276 | 0.203 |
| ZeroDiff | **0.596** | **1.423** | 0.736 | **0.846** | **3.109** | **2.494** | **0.868** | **1.756** | **1.399** | **0.788** | **0.219** | **0.159** |

† NSE < 0 (cross-location transfer failed). $f_\omega$: LSTM (Kratzert et al., 2018); + $f_\omega$ = pre-trained on $f_\omega$ predictions.

bidir. = bidirectional denoiser (§ 4.2); $w_k$ = moment-guided weighting (Eq. 12).

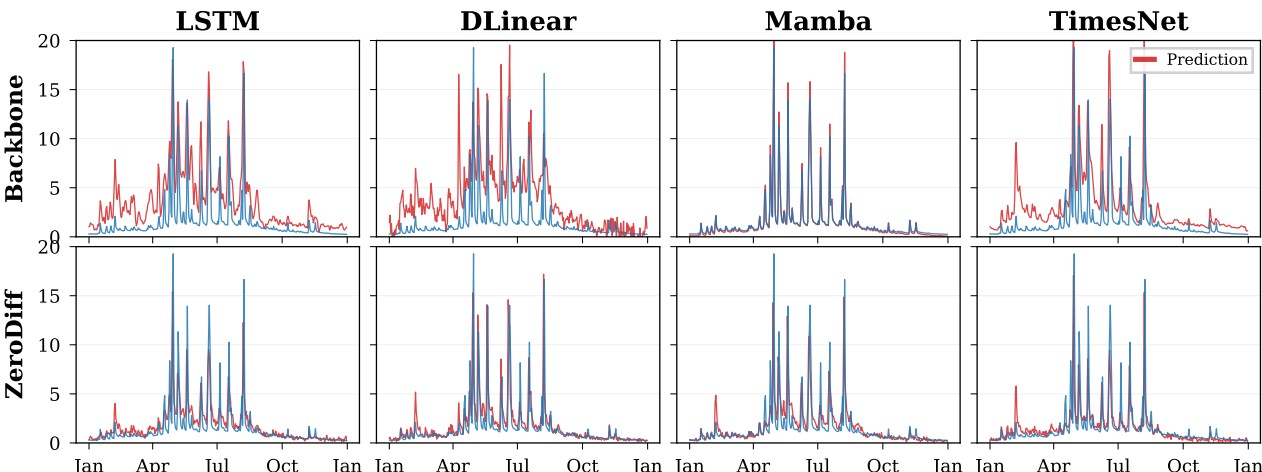

*Figure 3.* Qualitative comparison on Basin 02065500 (Year 1997, 365 days). Top row: reconstruction by each backbone dynamics model alone; bottom row: reconstruction enhanced by ZeroDiff. Blue: ground truth streamflow; red: model prediction. ZeroDiff consistently improves temporal fidelity, particularly at peak flow events.

full sequence. Final predictions are concatenated. Second, exogenous-only input assumption: For methods supporting exogenous variables, we provide $\mathbf{Y}_{1:L}$ reconstructed from $\mathbf{X}_{1:L}$ as input, matching our zero-shot setting. For each baseline, we report the better of the two protocols, selected on a held-out validation set. Further details on experimental fairness are provided in Appendix D.

## 5.2. Main Experiments

**Baselines.** Since no existing method directly addresses zero-shot time series reconstruction, we compare against representative diffusion-based approaches from two related tasks. From *time series imputation*: CSDI (Tashiro et al., 2021), which performs conditional score-based diffusion with self-supervised masking; SSSD (Alcaraz & Strodthoff, 2022), which combines structured state-space

*Table 2.* Effect of dynamics model architecture on CAMELS. Upper: dynamics model $f_\omega$ with moment estimation; lower: full ZeroDiff pipeline with each $f_\omega$.

| Model | NSE↑ | RMSE↓ | MAE↓ |
|---|---|---|---|
| *Dynamics model $f_\omega$ only* | | | |
| LSTM (Kratzert et al., 2018) | 0.481 | 1.679 | 0.725 |
| DLinear (Zeng et al., 2023) | 0.428 | 1.777 | 0.818 |
| Mamba (Gu & Dao, 2023) | 0.557 | 1.869 | 0.892 |
| TimesNet (Wu et al., 2023) | 0.517 | 1.613 | 0.706 |
| *ZeroDiff with $f_\omega$* | | | |
| ZeroDiff (LSTM) | 0.596 | 1.423 | 0.736 |
| ZeroDiff (DLinear) | 0.612 | 1.353 | 0.722 |
| ZeroDiff (Mamba) | 0.667 | 1.618 | 0.828 |
| ZeroDiff (TimesNet) | 0.593 | 1.427 | 0.724 |

models with diffusion for long-range temporal modeling; and CSBI (Chen et al., 2023), a Schrödinger bridge-based method for conditional generation. From *time series forecasting*: Diffusion-TS (Yuan & Qiao, 2024), which uses an encoder-decoder Transformer with disentangled seasonal-trend representations; and NsDiff (Ye et al., 2025), which introduces non-stationary diffusion to handle distribution shift. All baselines require target observations during training; we adapt them using the sliding-window and exogenous-conditioned protocols described in Appendix D, selecting the best configuration per baseline via validation.

**Results.** Table 1 shows a clear pattern: standard diffusion models fail (†) on Streamflow and Methane—datasets exhibiting high variability both across locations and within each time series. Without moment estimation to capture location-specific statistics, models trained on observed locations cannot generalize to unseen ones.

Providing baselines with $f_\omega$ predictions (rows + $f_\omega$) reduces errors. However, these models can only incorporate the prior through pre-training—using $f_\omega(\mathbf{X})$ as pseudo-labels. This treats the prior as training data rather than as an integral part of the diffusion process.

ZeroDiff instead integrates the informed prior directly into the forward process, reverse process, and optimization. The consistent gap between ZeroDiff and baselines (+ $f_\omega$) across all datasets demonstrates that tightly coupling the prior with diffusion is essential.

**Ablation Study.** We ablate ZeroDiff in the lower block of Table 1; effects are most pronounced on Streamflow where magnitudes differ by orders of magnitude across basins. Replacing $\mathcal{N}(\hat{\mathbf{Y}}, \sigma^2\mathbf{I})$ with $\mathcal{N}(\mathbf{0}, \mathbf{I})$ causes fail-

ure on Streamflow and Methane, while the deterministic prior $\Phi_{\psi,\omega}$ alone yields positive NSE everywhere; comparing $\Phi_{\psi,\omega}$ with ZeroDiff shows that diffusion calibration further improves performance by correcting systematic errors from dynamics learning. Removing the VAE collapses performance on high-variability datasets, as the shared dynamics model cannot rescale predictions without estimated $(\hat{\mu}, \hat{\sigma})$. Moment-guided weighting $w_k$ provides gains only when cross-location heterogeneity is high—on Solar and Temp where locations cluster in moment space, weighting becomes redundant. The bidirectional denoiser yields consistent improvement across all datasets: unlike forecasting where causality prohibits looking ahead, reconstruction allows information to flow both directions, enabling correction at sharp transitions.

### 5.3. Choice of Dynamics Model

The dynamics model $f_\omega$ is a plug-in component of ZeroDiff. In Table 2, we evaluate four architectures on CAMELS (Streamflow), the most challenging dataset: LSTM (Hochreiter & Schmidhuber, 1997), DLinear (Zeng et al., 2023), Mamba (Gu & Dao, 2023), and TimesNet (Wu et al., 2023).

As shown in Table 2, ZeroDiff consistently improves all four backbones, confirming that the diffusion refinement is architecture-agnostic. Notably, although Mamba achieves the highest standalone NSE (0.557), our default choice is the simplest LSTM—yet ZeroDiff (LSTM) already reaches 0.596, demonstrating that a correct formulation matters more than a powerful backbone. Even a naive dynamics model can capture sufficient temporal structure to provide a good prior for diffusion refinement. This also reveals the potential of ZeroDiff as a bridge between classical time series models and diffusion-based generation: practitioners can plug in any off-the-shelf backbone and obtain calibrated reconstructions. Figure 3 illustrates this qualitatively—across all four backbones, ZeroDiff tracks ground-truth peaks more faithfully than the backbone alone.

### 5.4. Robustness to Prior Estimation Error

The deterministic prior $\Phi_{\psi,\omega}$ varies substantially in quality across datasets and locations (Table 1), and the reverse posterior mean (Eq. 10) anchors the diffusion trajectory to this prior through the $\gamma_2$ term. We analyze how the final reconstruction depends on prior quality from three angles: the role of the VAE, the behavior of diffusion under prior error, and the mechanism that decouples the two stages.

**Decoupling moment estimation.** Zero-shot spatial extrapolation requires estimating both the marginal moments $(\mu, \sigma)$ and the temporal dynamics. The VAE decouples these two estimands, letting each component focus its generalization on one aspect. Table 4 shows that on low-heterogeneity

*Table 4.* Ablation of VAE moment estimation. NSE across four datasets.

| Pipeline | Solar | Temp | Streamflow | Methane |
|---|---|---|---|---|
| VAE + LSTM + Diffusion | 0.846 | 0.868 | 0.596 | 0.788 |
| LSTM + Diffusion (no VAE) | 0.839 | 0.828 | 0.377 | $-0.655$ |

*Table 5.* Prior estimation error and diffusion recovery. PRIOR NSE<0: locations where the deterministic prior fails. IMPROVED: locations where diffusion increases NSE.

| Dataset | Max/Min ratio | VAE error | Prior NSE<0 | Improved |
|---|---|---|---|---|
| Solar | $1.3\times$ | 2.0% | 0 | 32/32 |
| Temp | $2.0\times$ | 14.1% | 1 | 33/43 |
| Streamflow | $791\times$ | 18.9% | 1 | 89/98 |
| Methane | $728\times$ | 9.9% | 4 | 28/31 |

*Table 6.* Providing VAE-estimated moments $(\hat{\mu}, \hat{\sigma})$ as input features to the denoiser. Removing this conditioning consistently degrades performance.

| Dataset | $(\hat{\mu}, \hat{\sigma})$ as input | $(\hat{\mu}, \hat{\sigma})$ removed |
|---|---|---|
| Solar | 0.846 | 0.831 |
| Temp | 0.868 | 0.842 |
| Streamflow | 0.596 | 0.551 |
| Methane | 0.788 | 0.731 |

datasets (Solar, Temp), removing the VAE causes negligible change, but on high-heterogeneity datasets it causes severe degradation, collapsing to $-0.655$ NSE on Methane. An imperfect prior is far better than no prior.

**Diffusion recovery from prior error.** The $\gamma_2$ anchoring term ties the reverse process to $\hat{\mathbf{Y}}$, raising the possibility that inaccurate moments propagate into the final reconstruction. Empirically, the opposite holds: the calibration gain is largest precisely where the prior is weakest. Comparing the $\Phi_{\psi,\omega}$ and ZeroDiff rows of Table 1, the absolute NSE improvement on Methane is $+0.307$ ($0.481 \rightarrow 0.788$), the largest among the four datasets. Table 5 confirms this at the location level: even on Methane, all 4 locations with negative prior NSE are recovered to positive NSE after diffusion, and 28 of 31 locations overall see improvement.

**Moment-conditioned correction.** The VAE-estimated moments $(\hat{\mu}, \hat{\sigma})$ are not only used to construct $\hat{\mathbf{Y}}$ via denormalization—they are also explicitly provided as input features to the denoiser: $\epsilon_\theta = f_\theta(\mathbf{Y}_t, [\mathbf{X}; \hat{\mu}; \hat{\sigma}], \hat{\mathbf{Y}}, t)$. During training, the denoiser observes a spectrum of $(\hat{\mu}, \hat{\sigma})$ values across observed locations, each paired with the corresponding ground-truth $\mathbf{Y}$, and learns a correction policy parameterized by these moments rather than a fixed offset

*Table 7.* Multi-Target reconstruction (NSE). Each row holds out a different number of locations per fold; a single model is trained to reconstruct all held-out targets simultaneously.

| Locations out | Streamflow | Solar | Temp | Methane |
|---|---|---|---|---|
| 1 | 0.596 | 0.846 | 0.868 | 0.788 |
| 5 | 0.551 | 0.843 | 0.875 | 0.735 |
| 10 | 0.554 | 0.843 | 0.865 | 0.749 |
| 20 | 0.550 | 0.847 | 0.855 | 0.609 |

relative to any specific prior. Table 6 isolates this effect: removing $(\hat{\mu}, \hat{\sigma})$ from the denoiser's input consistently drops NSE, with the largest drops on high-heterogeneity datasets (Streamflow $-0.045$, Methane $-0.057$). Since the prior is constructed identically at observed and unobserved locations, this correction transfers to $\mathcal{U}$.

### 5.5. Multi-Target Reconstruction

In practical deployments such as continental-scale hydrological modeling, the set $\mathcal{U}$ of unobserved locations is large, and training a separate model per target is infeasible. ZeroDiff is designed to serve all targets with a single model: the moment-guided weighting (Eq. 12) uses the centroid $(\hat{\mu}^*, \hat{\sigma}^*)$ of estimated moments over $\mathcal{U}$ (Appendix E.1), up-weighting observed locations whose distributional characteristics are representative of the collective target set.

Table 7 varies the number of held-out locations per fold from 1 to 20. On low-heterogeneity datasets (Solar, Temp), performance is essentially unchanged as $|\mathcal{U}|$ grows, indicating that the centroid is an effective summary when targets cluster in moment space. On high-heterogeneity datasets, NSE drops modestly (Streamflow $0.596 \rightarrow 0.550$, Methane $0.788 \rightarrow 0.609$) as the centroid must compromise across more diverse targets. This degradation is a deliberate trade-off: one trained model serves all $|\mathcal{U}|$ locations, providing an $|\mathcal{U}|$-fold reduction in computational cost relative to per-location training, with the per-location accuracy gap remaining small when targets share a geographic region.

## 6. Conclusion

ZeroDiff enables zero-shot time series reconstruction by moment estimation and dynamics learning, constructing an informed prior from exogenous variables alone, then calibrating systematic errors through diffusion. The calibration learned on observed locations generalizes to unobserved ones. Experiments across four real-world datasets show that ZeroDiff significantly outperforms existing diffusion methods, particularly on datasets with high cross-location variability where baselines fail entirely.

## Acknowledgments

This research is supported by Dan Lu's Early Career Project, sponsored by the Office of Biological and Environmental Research in the U.S. Department of Energy (DOE). X.J. was partially supported by the National Science Foundation (NSF) grants 2203581, 2239175, 2316305, 2147195, 2425845, and 2530609; the USGS award G22AC00266; and the NASA grants 80NSSC24K1061 and 80NSSC25K0013.

## Impact Statement

This paper presents ZeroDiff for zero-shot time series reconstruction, with direct applications in hydrological modeling. Potential positive societal impacts include improved flood early warning, water resource management, and environmental monitoring in ungauged regions where monitoring infrastructure is limited. As with any predictive model, we encourage practitioners to appropriately communicate uncertainty estimates when informing real-world decisions.

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

# A. Derivation of Posterior Distribution

We derive the posterior distribution for our informed-prior diffusion process. For notational simplicity, we omit the location index and write $\mathbf{Y}$ for the target, $\hat{\mathbf{Y}}$ for the informed prior, and $\mathbf{X}$ for the exogenous input. Unlike standard DDPM which diffuses toward pure noise $\mathbf{Y}_T \sim \mathcal{N}(\mathbf{0}, \mathbf{I})$, we diffuse toward the informed prior $\hat{\mathbf{Y}}$, enabling warm-start initialization for calibration.

## A.1. Forward Process

The one-step forward transition is:

$$\mathbf{Y}_t = \sqrt{\alpha_t}\mathbf{Y}_{t-1} + (1 - \sqrt{\alpha_t})\hat{\mathbf{Y}} + \sqrt{\beta_t}\boldsymbol{\epsilon}_t, \quad \boldsymbol{\epsilon}_t \sim \mathcal{N}(\mathbf{0}, \mathbf{I}) \tag{15}$$

where $\alpha_t := 1 - \beta_t$, $\bar{\alpha}_t := \prod_{i=1}^{t} \alpha_i$, and $\bar{\sigma}_t := 1 - \bar{\alpha}_t$. Unrolling recursively:

$$
\begin{aligned}
\mathbf{Y}_t &= \sqrt{\alpha_t}\mathbf{Y}_{t-1} + (1 - \sqrt{\alpha_t})\hat{\mathbf{Y}} + \sqrt{\beta_t}\boldsymbol{\epsilon}_t \\
&= \sqrt{\alpha_t}\left[\sqrt{\alpha_{t-1}}\mathbf{Y}_{t-2} + (1 - \sqrt{\alpha_{t-1}})\hat{\mathbf{Y}} + \sqrt{\beta_{t-1}}\boldsymbol{\epsilon}_{t-1}\right] + (1 - \sqrt{\alpha_t})\hat{\mathbf{Y}} + \sqrt{\beta_t}\boldsymbol{\epsilon}_t \\
&= \sqrt{\alpha_t\alpha_{t-1}}\mathbf{Y}_{t-2} + \left(1 - \sqrt{\alpha_t\alpha_{t-1}}\right)\hat{\mathbf{Y}} + \sqrt{\alpha_t\beta_{t-1} + \beta_t}\boldsymbol{\epsilon} \\
&\vdots \\
&= \sqrt{\bar{\alpha}_t}\mathbf{Y} + (1 - \sqrt{\bar{\alpha}_t})\hat{\mathbf{Y}} + \sqrt{\bar{\sigma}_t}\boldsymbol{\epsilon}
\end{aligned}
\tag{16}
$$

Thus $q(\mathbf{Y}_t|\mathbf{Y}, \hat{\mathbf{Y}}) = \mathcal{N}\left(\sqrt{\bar{\alpha}_t}\mathbf{Y} + (1 - \sqrt{\bar{\alpha}_t})\hat{\mathbf{Y}}, \bar{\sigma}_t\mathbf{I}\right)$, and at $t = T$: $\mathbf{Y}_T \sim \mathcal{N}(\hat{\mathbf{Y}}, \bar{\sigma}_T\mathbf{I})$—the warm-start initialization centered at the prior.

## A.2. Reverse Posterior Derivation

We derive $q(\mathbf{Y}_{t-1}|\mathbf{Y}_t, \mathbf{Y}, \hat{\mathbf{Y}})$ via Bayes' rule:

$$q(\mathbf{Y}_{t-1}|\mathbf{Y}_t, \mathbf{Y}, \hat{\mathbf{Y}}) \propto q(\mathbf{Y}_t|\mathbf{Y}_{t-1}, \hat{\mathbf{Y}}) \cdot q(\mathbf{Y}_{t-1}|\mathbf{Y}, \hat{\mathbf{Y}}) \tag{17}$$

From Eq. (15), the transition density is:

$$q(\mathbf{Y}_t|\mathbf{Y}_{t-1}, \hat{\mathbf{Y}}) = \mathcal{N}\left(\sqrt{\alpha_t}\mathbf{Y}_{t-1} + (1 - \sqrt{\alpha_t})\hat{\mathbf{Y}}, \beta_t\mathbf{I}\right) \tag{18}$$

From Eq. (16) at $t - 1$:

$$q(\mathbf{Y}_{t-1}|\mathbf{Y}, \hat{\mathbf{Y}}) = \mathcal{N}\left(\sqrt{\bar{\alpha}_{t-1}}\mathbf{Y} + (1 - \sqrt{\bar{\alpha}_{t-1}})\hat{\mathbf{Y}}, \bar{\sigma}_{t-1}\mathbf{I}\right) \tag{19}$$

Define $\mathbf{A} := \mathbf{Y}_t - (1 - \sqrt{\alpha_t})\hat{\mathbf{Y}}$ and $\mathbf{B} := \sqrt{\bar{\alpha}_{t-1}}\mathbf{Y} + (1 - \sqrt{\bar{\alpha}_{t-1}})\hat{\mathbf{Y}}$. The product of two Gaussians gives:

$$
\begin{aligned}
q(\mathbf{Y}_{t-1}|\mathbf{Y}_t, \mathbf{Y}, \hat{\mathbf{Y}}) &\propto \exp\left(-\frac{(\mathbf{A} - \sqrt{\alpha_t}\mathbf{Y}_{t-1})^2}{2\beta_t} - \frac{(\mathbf{Y}_{t-1} - \mathbf{B})^2}{2\bar{\sigma}_{t-1}}\right) \\
&= \exp\left(-\frac{1}{2}\left[\left(\frac{\alpha_t}{\beta_t} + \frac{1}{\bar{\sigma}_{t-1}}\right)\mathbf{Y}_{t-1}^2 - 2\left(\frac{\sqrt{\alpha_t}\mathbf{A}}{\beta_t} + \frac{\mathbf{B}}{\bar{\sigma}_{t-1}}\right)\mathbf{Y}_{t-1} + \text{const}\right]\right)
\end{aligned}
\tag{20}
$$

Completing the square in $\mathbf{Y}_{t-1}$, the posterior is Gaussian with:

$$\tilde{\sigma}_{t-1} = \left(\frac{\alpha_t}{\beta_t} + \frac{1}{\bar{\sigma}_{t-1}}\right)^{-1} = \frac{\beta_t\bar{\sigma}_{t-1}}{\alpha_t\bar{\sigma}_{t-1} + \beta_t} \tag{21}$$

$$\tilde{\mu}_{t-1} = \tilde{\sigma}_{t-1}\left(\frac{\sqrt{\alpha_t}\mathbf{A}}{\beta_t} + \frac{\mathbf{B}}{\bar{\sigma}_{t-1}}\right) = \frac{\sqrt{\alpha_t}\bar{\sigma}_{t-1}\mathbf{A} + \beta_t\mathbf{B}}{\alpha_t\bar{\sigma}_{t-1} + \beta_t} \tag{22}$$

Substituting $\mathbf{A} = \mathbf{Y}_t - (1 - \sqrt{\alpha_t})\hat{\mathbf{Y}}$ and $\mathbf{B} = \sqrt{\bar{\alpha}_{t-1}}\mathbf{Y} + (1 - \sqrt{\bar{\alpha}_{t-1}})\hat{\mathbf{Y}}$ into Eq. (22):

$$
\begin{aligned}
\tilde{\mu}_{t-1} &= \frac{\sqrt{\alpha_t}\bar{\sigma}_{t-1}\left[\mathbf{Y}_t - (1 - \sqrt{\alpha_t})\hat{\mathbf{Y}}\right] + \beta_t\left[\sqrt{\bar{\alpha}_{t-1}}\mathbf{Y} + (1 - \sqrt{\bar{\alpha}_{t-1}})\hat{\mathbf{Y}}\right]}{\alpha_t\bar{\sigma}_{t-1} + \beta_t} \\
&= \frac{\sqrt{\bar{\alpha}_{t-1}}\beta_t}{\alpha_t\bar{\sigma}_{t-1} + \beta_t}\mathbf{Y} + \frac{\sqrt{\alpha_t}\bar{\sigma}_{t-1}}{\alpha_t\bar{\sigma}_{t-1} + \beta_t}\mathbf{Y}_t + \frac{\sqrt{\alpha_t}(\sqrt{\alpha_t} - 1)\bar{\sigma}_{t-1} + (1 - \sqrt{\bar{\alpha}_{t-1}})\beta_t}{\alpha_t\bar{\sigma}_{t-1} + \beta_t}\hat{\mathbf{Y}}
\end{aligned}
\tag{23}
$$

Thus the posterior mean has the **three-term form**:

$$
\boxed{\tilde{\mu}_{t-1} = \gamma_0\mathbf{Y} + \gamma_1\mathbf{Y}_t + \gamma_2\hat{\mathbf{Y}}}
\tag{24}
$$

where:

$$
\gamma_0 = \frac{\sqrt{\bar{\alpha}_{t-1}}\beta_t}{\alpha_t\bar{\sigma}_{t-1} + \beta_t}, \quad \gamma_1 = \frac{\sqrt{\alpha_t}\bar{\sigma}_{t-1}}{\alpha_t\bar{\sigma}_{t-1} + \beta_t}, \quad \gamma_2 = \frac{\sqrt{\alpha_t}(\sqrt{\alpha_t} - 1)\bar{\sigma}_{t-1} + (1 - \sqrt{\bar{\alpha}_{t-1}})\beta_t}{\alpha_t\bar{\sigma}_{t-1} + \beta_t}
\tag{25}
$$

This differs from standard DDPM where $\tilde{\mu}_{t-1}^{\text{DDPM}} = \gamma_0\mathbf{Y} + \gamma_1\mathbf{Y}_t$. The additional $\gamma_2\hat{\mathbf{Y}}$ term anchors each denoising step to the prior prediction.

### A.3. Inference via Reparameterization

During inference, $\mathbf{Y}$ is unknown. From Eq. (16), we estimate:

$$
\hat{\mathbf{Y}}_0 = \frac{1}{\sqrt{\bar{\alpha}_t}}\left(\mathbf{Y}_t - (1 - \sqrt{\bar{\alpha}_t})\hat{\mathbf{Y}} - \sqrt{\bar{\sigma}_t}\epsilon_\theta(\mathbf{Y}_t, \mathbf{X}, \hat{\mathbf{Y}}, t)\right)
\tag{26}
$$

where $\epsilon_\theta$ is the learned denoiser. The reverse step is:

$$
\mathbf{Y}_{t-1} = \gamma_0\hat{\mathbf{Y}}_0 + \gamma_1\mathbf{Y}_t + \gamma_2\hat{\mathbf{Y}} + \sqrt{\tilde{\sigma}_{t-1}}\mathbf{z}, \quad \mathbf{z} \sim \mathcal{N}(\mathbf{0}, \mathbf{I})
\tag{27}
$$

The prior $\hat{\mathbf{Y}}$ anchors the process at three levels: (1) initialization $\mathbf{Y}_T \sim \mathcal{N}(\hat{\mathbf{Y}}, \bar{\sigma}_T\mathbf{I})$, (2) reparameterization of $\hat{\mathbf{Y}}_0$, and (3) the $\gamma_2\hat{\mathbf{Y}}$ term in the posterior mean. This triple anchoring transforms diffusion from generation to calibration: the network learns residual corrections rather than the full data distribution.

# B. Training and Inference

---

**Algorithm 1** Training

---

**Input:** $\mathcal{O}, \mathcal{U}, \{\mathbf{X}^{(k)}\}, \{\mathbf{Y}^{(k)}\}_{k \in \mathcal{O}}$, diffusion steps $T$
Train VAE on $\mathcal{O}$: encoder $q_\psi$, feature encoder $\phi$, decoder $\mathrm{Dec}_\psi$         *// cross-modal moment estimation*
For all $k$: $(\hat{\mu}_k, \hat{\sigma}_k) \leftarrow \mathrm{Dec}_\psi(\mathbf{0}, \phi(\mathbf{Z}^{(k)}))$         *// infer magnitude from* $\mathbf{X}$
Train dynamics model $f_\omega$ on $\mathcal{O}$ in standardized space         *// shared response pattern*
Compute $(\hat{\mu}^*, \hat{\sigma}^*) \leftarrow \frac{1}{|\mathcal{U}|} \sum_{j \in \mathcal{U}} (\hat{\mu}_j, \hat{\sigma}_j)$         *// target centroid in moment space*
**repeat**
     Sample $k \in \mathcal{O}$ with probability $\propto w_k$         *// moment-guided weighting*
     Compute $\hat{\mathbf{Y}}^{(k)} \leftarrow \hat{\mu}_k + \hat{\sigma}_k \cdot f_\omega(\mathbf{X}^{(k)})$         *// pattern + magnitude*
     Sample $t \sim \mathrm{Uniform}(\{1, \ldots, T\})$, $\boldsymbol{\epsilon} \sim \mathcal{N}(\mathbf{0}, \mathbf{I})$
     $\mathbf{Y}_t^{(k)} \leftarrow \sqrt{\bar{\alpha}_t}\mathbf{Y}^{(k)} + (1 - \sqrt{\bar{\alpha}_t})\hat{\mathbf{Y}}^{(k)} + \sqrt{\bar{\sigma}_t}\boldsymbol{\epsilon}$         *// diffuse toward prior*
     Update $\theta$ via $\nabla_\theta w_k \|\boldsymbol{\epsilon} - \boldsymbol{\epsilon}_\theta(\mathbf{Y}_t^{(k)}, \mathbf{X}^{(k)}, \hat{\mathbf{Y}}^{(k)}, t)\|^2$         *// calibration on* $\mathcal{O}$
**until** convergence

---

---

**Algorithm 2** Inference

---

**Input:** $j \in \mathcal{U}$, $\mathbf{X}^{(j)}$, trained VAE ($\phi, \mathrm{Dec}_\psi$), dynamics model $f_\omega$, denoiser $\boldsymbol{\epsilon}_\theta$
$(\hat{\mu}_j, \hat{\sigma}_j) \leftarrow \mathrm{Dec}_\psi(\mathbf{0}, \phi(\mathbf{Z}^{(j)}))$         *// zero-shot moment estimation*
$\hat{\mathbf{Y}}^{(j)} \leftarrow \hat{\mu}_j + \hat{\sigma}_j \cdot f_\omega(\mathbf{X}^{(j)})$         *// pattern + magnitude*
$\mathbf{Y}_T^{(j)} \sim \mathcal{N}(\hat{\mathbf{Y}}^{(j)}, \bar{\sigma}_T \mathbf{I})$         *// warm-start from prior, not noise*
**for** $t = T$ **to** $1$ **do**
     $\boldsymbol{\epsilon}_\theta \leftarrow f_\theta(\mathbf{Y}_t^{(j)}, \mathbf{X}^{(j)}, \hat{\mathbf{Y}}^{(j)}, t)$         *// bidirectional conditioning*
     $\hat{\mathbf{Y}}_0^{(j)} \leftarrow \frac{1}{\sqrt{\bar{\alpha}_t}}\left(\mathbf{Y}_t^{(j)} - (1 - \sqrt{\bar{\alpha}_t})\hat{\mathbf{Y}}^{(j)} - \sqrt{\bar{\sigma}_t}\boldsymbol{\epsilon}_\theta\right)$         *// reparameterization*
     $\boldsymbol{\mu}_{t-1}^{(j)} \leftarrow \gamma_0 \hat{\mathbf{Y}}_0^{(j)} + \gamma_1 \mathbf{Y}_t^{(j)} + \gamma_2 \hat{\mathbf{Y}}^{(j)}$         *// three-term posterior*
     $\mathbf{Y}_{t-1}^{(j)} \leftarrow \boldsymbol{\mu}_{t-1}^{(j)} + \sigma_t \mathbf{z}, \quad \mathbf{z} \sim \mathcal{N}(\mathbf{0}, \mathbf{I})$ if $t > 1$
**end for**
**Output:** $\mathbf{Y}_0^{(j)}$

---

**Remark.** Unlike standard DDPM that initializes from $\mathcal{N}(\mathbf{0}, \mathbf{I})$, our method performs warm-start diffusion from the informed prior $\hat{\mathbf{Y}}^{(j)}$. The prior anchors the reverse process at three levels: (i) initialization of $\mathbf{Y}_T^{(j)}$, (ii) the reparameterization of $\hat{\mathbf{Y}}_0^{(j)}$, and (iii) the $\gamma_2$ term in the posterior mean. This transforms diffusion from generation to calibration—refining a reasonable estimate rather than constructing from noise.

# C. Dataset Details

We describe the four datasets, their exogenous variables, and train/test splits.

## C.1. Dataset Overview

*Table 8.* Dataset statistics: temporal coverage, train/test splits, and exogenous variable counts. All datasets use daily resolution.

| DATASET | FULL PERIOD | TRAIN PERIOD | DAYS | TEST LOC. | DYNAMIC | STATIC | EXOG. |
|---|---|---|---|---|---|---|---|
| CAMELS | 1989–2009 | 1989–2001 | 4,745 | 100 | 6 | 27 | 33 |
| FlowTemp | 2010–2012 | 2010–2012 | 1,095 | 42 | 8 | 0 | 8 |
| Solar | 1994–2007 | 1994–2004 | 4,015 | 30 | 45 | 3 | 48 |
| Mathane | 2008–2018 | 2008–2018 | 4,015 | 30 | 7 | 9 | 16 |

## C.2. Dataset Partition

The data splitting strategy for zero-shot reconstruction differs fundamentally from time series forecasting. While forecasting typically partitions data along the temporal dimension, our task requires spatial partitioning: for each experiment, we mask all target observations at a held-out location while retaining its exogenous variables, then leverage data from remaining locations to reconstruct the masked targets.

We reserve a subset of locations as the validation set. Although their targets are masked during training, we use their reconstruction performance to tune hyperparameters. The remaining held-out locations form the test set for final evaluation. This spatial cross-validation ensures that hyperparameters are selected without leaking information from test locations.

For most baselines, we do not adopt the default hyperparameters reported in their original papers, as these were optimized for forecasting tasks. Instead, we conduct task-specific hyperparameter search on our validation set to ensure fair comparison.

**Leave-One-Location-Out Evaluation.** We adopt a leave-one-location-out protocol to evaluate zero-shot generalization. For each dataset, we iteratively hold out one target location: all target observations at this location are masked while exogenous variables remain available. The model is trained on the remaining locations and evaluated on its ability to reconstruct the held-out target series. This procedure is repeated for all target locations—100 times for CAMELS, 42 for NHD, and 30 for both Solar and Methane. Final metrics are averaged across all held-out locations to provide a robust estimate of zero-shot reconstruction performance.

# D. Fairness Details

Most baselines are designed for time series forecasting with look-back window $L_b$ and prediction horizon $H$. To fairly evaluate on reconstruction ($L = 365$), we adopt two protocols.

### D.1. Sliding window reconstruction

We partition the sequence into segments where baselines use ground-truth $\mathbf{Y}$ from observed locations as look-back to predict subsequent horizons. We consider $L_b, H \in \{96, 192, 365\}$, including the $L_b = H = 365$ setting where the model reconstructs the current window conditioned on itself as a calibration baseline. Predictions are concatenated with stride $H$ to cover the full sequence. The optimal $(L_b, H)$ for each baseline is selected via validation set performance and reported in Table 1.

### D.2. Exogenous-conditioned reconstruction.

For methods supporting exogenous inputs, we provide the informed prior $\hat{\mathbf{Y}}_{1:L}$ reconstructed from $\mathbf{X}_{1:L}$ as the look-back window, matching our zero-shot setting where no ground-truth targets are available. This protocol evaluates whether baselines can refine an exogenous-based estimate.

Note that protocol (i) provides baselines with oracle access to target observations—an advantage unavailable in true zero-shot scenarios.

### D.3. Hyperparameter Tuning

For zero-shot time series reconstruction, dataset splitting differs fundamentally from time series forecasting. While forecasting typically partitions data along the temporal dimension, our task partitions along the spatial dimension. In each experiment, we mask all target observations at a held-out location while retaining its exogenous variables, then leverage data from other locations to perform zero-shot prediction of the masked target variable. We reserve several locations as a validation set; although their targets are masked during inference, we use their ground-truth values for hyperparameter selection.

For most baselines, we do not use the default hyperparameters reported in their original papers, as those were tuned for time series forecasting rather than our reconstruction task. Instead, we conduct task-specific hyperparameter search. Our experiments reveal that among $L_b, H \in \{96, 192, 365\}$, the configuration $L_b = H = 365$ performs best for most baselines—reconstructing a full year from a full year of context. We attribute this to the misalignment artifacts that arise when $L_b$ or $H$ take smaller values, as discontinuous segments must be concatenated to cover the annual sequence.

# E. Efficiency

## E.1. Multi-Target Reconstruction

A natural question arises: why not train a separate model for each target location $j \in \mathcal{U}$? Location-specific training would allow moment-guided weighting to focus entirely on $(\hat{\mu}_j, \hat{\sigma}_j)$, potentially yielding better reconstruction for that specific location.

While this approach maximizes per-location performance, it becomes impractical when $|\mathcal{U}|$ is large. Training and storing separate models for each target location incurs computational cost linear in $|\mathcal{U}|$—prohibitive in applications such as continental-scale hydrological modeling where thousands of ungauged locations require reconstruction.

Our framework addresses this via the moment centroid. Let $(\hat{\mu}^*, \hat{\sigma}^*)$ denote the centroid of estimated statistics over $j \in \mathcal{U}$:

$$(\hat{\mu}^*, \hat{\sigma}^*) = \frac{1}{|\mathcal{U}|} \sum_{j \in \mathcal{U}} (\hat{\mu}_j, \hat{\sigma}_j) \tag{28}$$

By defining $(\hat{\mu}^*, \hat{\sigma}^*)$ as the geometric center over all $j \in \mathcal{U}$, a single model serves multiple target locations simultaneously. The moment-guided weighting $w_k$ (Eq. 12) then upweights training locations whose distributions are representative of the *collective* target set, rather than any individual target.

This design reflects a deliberate trade-off: we accept a modest reduction in per-location accuracy in exchange for an $|\mathcal{U}|$-fold improvement in efficiency. In practice, when target locations cluster in moment space—as is common within a geographic region—the centroid closely approximates individual targets, and the performance gap is minimal. Table 7 in the main text validates this empirically: as $|\mathcal{U}|$ grows from 1 to 20, performance remains nearly unchanged on low-heterogeneity datasets (Solar, Temp) and degrades only modestly on high-heterogeneity datasets, while training cost is reduced by a factor of $|\mathcal{U}|$.

For applications requiring maximum accuracy at a specific location, one can set $\mathcal{U} = \{j\}$ and train a location-specific model. Our framework supports both modes: efficient batch reconstruction when throughput matters, and focused single-target reconstruction when accuracy is paramount.

## E.2. Diffusion Computational Efficiency Analysis

*Table 9.* Computational efficiency of six diffusion-based imputation models across all datasets. Models are sorted by total per-fold time (ascending) within each panel. All datasets use 365-day sequences. Training uses early stopping with patience of 20 epochs. All experiments conducted on H100.

| Model | #Params | $T$ | $N_s$ | Train (min) | Infer (min) | ms/sample | Total (min) |
|---|---|---|---|---|---|---|---|
| **(a) Streamflow** *(531 location, 33 features, 6,903 samples/split)* | | | | | | | |
| ZeroDiff | 271 K | 20 | 100 | **4.3** | 9.1 | 39.4 | **13.4** |
| DiffusionTS[†] | 2.33 M | 100 | 1 | 12.1 | **3.0** | **21.4** | 15.1 |
| NsDiff | 270 K | 20 | 100 | 16.8 | 9.2 | 39.9 | 26.0 |
| CSDI | 618 K | 50 | 10 | 23.0 | 44.4 | 192.9 | 67.4 |
| SSSD[‡] | 29.4 M | 200 | 10 | 12.2 | 165.8 | 720.5 | 178.0 |
| CSBI | 137 K | 100 | 20 | 19.1 | 160.0 | 695.4 | 179.1 |
| **(b) Solar** *(500 locations, 16 features, 5,500 samples/split)* | | | | | | | |
| DiffusionTS[†] | 2.32 M | 100 | 1 | 6.3 | **4.0** | **22.0** | **10.3** |
| ZeroDiff | 270 K | 20 | 100 | **4.2** | 7.2 | 39.5 | 11.4 |
| NsDiff | 269 K | 20 | 100 | 12.4 | 7.3 | 39.9 | 19.7 |
| CSDI | 617 K | 50 | 10 | 12.3 | 35.4 | 192.9 | 47.7 |
| CSBI | 137 K | 100 | 20 | 15.2 | 127.2 | 693.8 | 142.4 |
| SSSD[‡] | 29.4 M | 200 | 10 | 10.8 | 132.4 | 722.2 | 143.2 |
| **(c) Methane / Temp**[*] *(45 locations, 17 features, 135 samples/split)* | | | | | | | |
| ZeroDiff | 269 K | 20 | 100 | **0.6** | 0.5 | 77.0 | **1.1** |
| NsDiff | 269 K | 20 | 100 | 0.8 | 0.5 | 76.9 | 1.3 |
| DiffusionTS[†] | 2.32 M | 100 | 1 | 1.8 | **0.3** | **41.0** | 2.1 |
| CSDI | 617 K | 50 | 10 | 1.6 | 2.2 | 332.1 | 3.8 |
| SSSD[‡] | 29.4 M | 200 | 10 | 1.0 | 5.2 | 770.0 | 6.2 |
| CSBI | 137 K | 100 | 20 | 1.0 | 12.3 | 1,821.0 | 13.3 |

# F. Overall Visualization

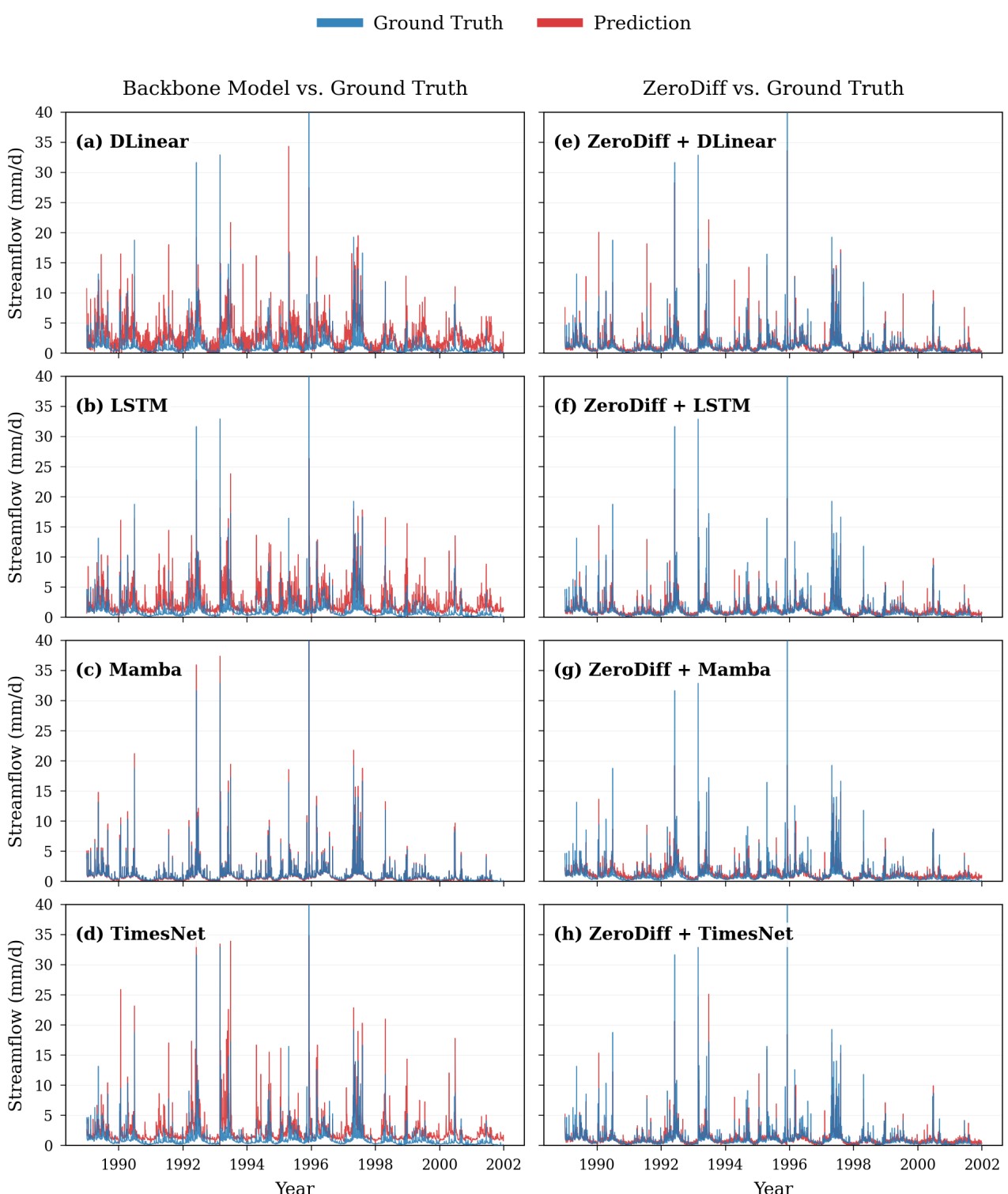

*Figure 4.* Full time series reconstruction for Basin 02065500 over the entire 13-year training period (1989–2001, 4,745 days).

## G. Hyperparameter Search

We conduct a systematic hyperparameter sensitivity analysis for the diffusion component of ZeroDiff across all four datasets. We vary one factor at a time while keeping the remaining settings at their defaults ($d_{\text{model}} = 512$, $n_{\text{layers}} = 2$, $T = 20$, lr $= 10^{-3}$, no fusion). Results are summarized in Figure 5.

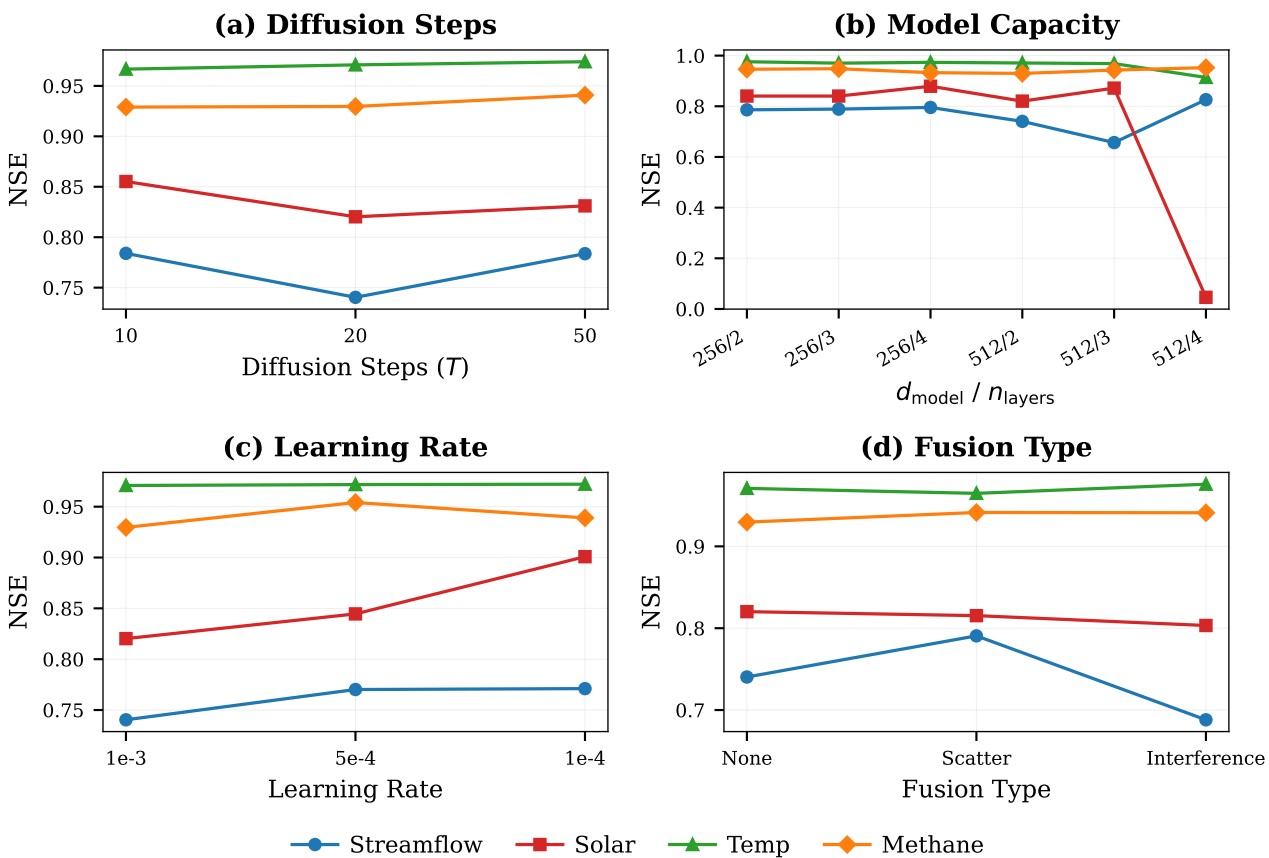

*Figure 5.* Hyperparameter sensitivity of ZeroDiff across four datasets, measured by NSE. Each panel varies one factor: (a) diffusion steps $T$, (b) denoiser capacity ($d_{\text{model}}$ / $n_{\text{layers}}$), (c) learning rate, and (d) exogenous fusion strategy.

**Diffusion steps.** Increasing $T$ from 10 to 50 yields modest but consistent gains on most datasets, with $T = 50$ achieving the best NSE on Streamflow, Temp, and Methane. The improvement is marginal relative to the added computational cost, so we adopt $T = 20$ as the default for efficiency.

**Model capacity.** Smaller denoisers ($d_{\text{model}} = 256$) perform comparably to or better than larger ones ($d_{\text{model}} = 512$) at moderate depth. However, over-parameterization can be harmful: the $512/4$ configuration causes a severe performance collapse on Solar (NSE $= 0.05$) and a notable drop on Temp, indicating overfitting when the denoiser capacity exceeds what the calibration task requires.

**Learning rate.** Performance is relatively stable across the tested range ($10^{-3}$ to $10^{-4}$). Solar benefits most from a lower learning rate ($10^{-4}$), while other datasets show mild improvements or remain flat. We use $10^{-3}$ as the default.

**Fusion type.** The exogenous fusion strategy has a dataset-dependent effect. Scatter fusion improves Streamflow and Methane, while interference fusion benefits Temp. No single fusion strategy dominates across all datasets, so we default to no fusion for simplicity and leave fusion selection as a dataset-specific tuning option.

