# OpenReview forum: "ZeroDiff: Zero-Shot Time Series Reconstruction via Informed-Prior Diffusion"
_ICML.cc/2026/Conference — ICML 2026 regular_

### Official Review · Reviewer_KfzK · 2026-03-03

**Soundness:** 2
**Presentation:** 2
**Significance:** 2
**Originality:** 2
**Overall Recommendation:** 2
**Confidence:** 4

**Summary:**

The paper deals with the zero-shot time series reconstruction issue. A new method ZeroDiff is proposed, which constructs an informed prior from exogenous variables. Some experiments are designed for comparison.

**Compliance With Llm Reviewing Policy:**

Affirmed.

**Key Questions For Authors:**

1) Without the assumption of Gaussianity, would the proposed approach work for the zero-shot time series reconstruction?
2) What are the merits of using the informed prior? What are the theoretical guarantees for the proposed approach?
3) For time series data, the extrapolation as described in the related work is a common method. Does the proposed method improve the extrapolation? Some numerical comparison needs to be done.

**Limitations:**

The proposed approach relies on a Gaussian prior, yet the manuscript does not provide formal theoretical guarantees that justify this modeling choice or characterize its impact on performance.
The empirical gains appear modest, making it difficult to assess the practical advantage of the proposed approach.
These factors limit the strength of the theoretical and empirical support for the method.

**Strengths And Weaknesses:**

Soundness:
The analysis makes use of well-known properties of the Gaussian distribution. It remains unclear under what conditions the method is guaranteed to perform well, or how its behavior compares to existing approaches from a theoretical standpoint. The reliance on Gaussian properties alone does not by itself constitute a rigorous justification of the method’s practical or theoretical advantages.

Presentation:
The manuscript does not clearly describe the concrete procedural steps required to implement the proposed method.
The absence of a clearly specified algorithmic workflow makes it difficult to fully understand and reproduce the method.
Providing an explicit algorithmic description or pseudocode would substantially improve the clarity of the presentation.

Significance:
The manuscript does not clearly establish the importance of the addressed problem within the broader literature. It remains unclear how central or impactful the proposed formulation is relative to existing approaches.

Originality:
The proposed method relies fundamentally on a Gaussian prior, a modeling choice that is well established in the literature. The manuscript does not clearly demonstrate how this usage leads to conceptually new theoretical insights or methodological innovations.

---

> ### Author Rebuttal · Authors · 2026-03-30
>
> We thank the reviewer for the insightful questions and address each below.
>
> **Q1.**
>
> The Gaussian diffusion framework is appropriate here because the informed prior absorbs the non-Gaussian structure of the target — what diffusion models is the calibration residual, which is approximately symmetric. We verify this by analyzing the calibration residual $\hat{\mathbf{Y}}_{\text{diff}} - \hat{\mathbf{Y}}$ on 25 held-out (zero-shot) locations (118,625 data points):
>
> *Table 1: Target distribution vs. calibration residual on zero-shot locations (Streamflow).*
>
> | Distribution | Mean | Median | Neg/Pos | Skewness | Std |
> |:--|:--:|:--:|:--:|:--:|:--:|
> | $Y$ (target) | 1.846 | 0.990 | 0/100% | 6.46 | 2.936 |
> | $\hat{\mathbf{Y}}_{\text{diff}} - \hat{\mathbf{Y}}$ (calibration residual) | -0.032 | -0.061 | 56/44% | 5.87 | 1.082 |
>
> *$\hat{Y}$: informed prior (Stage 1); $\hat{Y}_{\text{diff}}$: diffusion output.*
>
> The informed prior converts the problem from modeling a strictly positive, right-skewed distribution into learning a correction signal that is centered (mean reduced by $57\times$) and bidirectional (sign ratio: 0%/100% $\to$ 56%/44%). A flow-regime decomposition confirms the remaining skewness ($5.87$) is a tail artifact:
>
> *Table 2: Calibration residual statistics by flow regime — normal flow (90%) vs. extreme events (10%).*
>
> | Flow regime | Fraction | Skewness | Neg/Pos |
> |:--|:--:|:--:|:--:|
> | Normal flow ($Y \leq Q_{90}$) | 90% | 0.40 | 56/44% |
> | Extreme flow ($Y > Q_{90}$) | 10% | 2.74 | 53/47% |
>
> For 90% of timesteps, the calibration residual is symmetric (skewness $= 0.40$). The elevated overall skewness is driven by the 10% of extreme events, where corrections remain balanced between over- and under-estimation (53%/47%). The informed prior absorbs the non-Gaussian structure of $Y$, leaving diffusion with a well-centered, symmetric-in-bulk correction signal.
>
>
>
> **Q2.**
>
> ZeroDiff's forward process (Eq. 9) gives $q(\mathbf{y}_T \mid \mathbf{y}_0, \hat{\mathbf{Y}}) = \mathcal{N}(\sqrt{\bar{\alpha}_T}\mathbf{y}_0 + (1{-}\sqrt{\bar{\alpha}_T})\hat{\mathbf{Y}}, \bar{\sigma}_T\mathbf{I})$ and initializes the reverse from $p(\mathbf{y}_T) = \mathcal{N}(\hat{\mathbf{Y}}, \bar{\sigma}_T\mathbf{I})$. Applying the non-asymptotic $W_2$ convergence framework for conditional diffusion models (Li, 2025, Theorem 1):
>
> $W_2(q_0, p_0) \leq \int_0^T \beta_t M(t)\sqrt{H(t)}\,dt + M(T) \cdot W_2(q(\mathbf{y}_T \mid \hat{\mathbf{Y}}), p(\mathbf{y}_T))$
>
> Both $q(\mathbf{y}_T \mid \mathbf{y}_0, \hat{\mathbf{Y}})$ and $p(\mathbf{y}_T)$ share covariance $\bar{\sigma}_T\mathbf{I}$, since $\hat{\mathbf{Y}}$ serves as both the forward diffusion endpoint and the reverse initialization. By the Gaussian $W_2$ formula, $W_2^2$ reduces to the squared mean difference; taking the expectation over $\mathbf{y}_0$ via mixture convexity:
>
> $W_2(q_0, p_0) \leq \int_0^T \beta_t M(t)\sqrt{H(t)}\,dt + M(T)\sqrt{\bar{\alpha}_T} \cdot \sqrt{\mathbb{E}[\|\mathbf{y}_0 - \hat{\mathbf{Y}}\|^2]}$
>
> The initialization error is controlled by the prior reconstruction error
> $\mathbb{E}[\|\mathbf{y}_0 - \hat{\mathbf{Y}}\|^2]$.
> Setting $\hat{\mathbf{Y}} = \mathbf{0}$ (no informed prior) yields initialization error proportional to $\sqrt{\mathbb{E}[\|\mathbf{y}_0\|^2]}$.
> Any prior satisfying $\mathbb{E}[\|\mathbf{y}_0 - \hat{\mathbf{Y}}\|^2] < \mathbb{E}[\|\mathbf{y}_0\|^2]$
>
> tightens the bound. In Table 1, $\Phi_{\psi,\omega}$ achieves this on all four datasets; the w/o prior ablation reverts to $\hat{\mathbf{Y}} = \mathbf{0}$ and fails (NSE $<0$) on Streamflow and Methane, confirming the theory.
>
> *Ref: Li (2025), arXiv:2508.10944.*
>
>
> **Q3.**
>
> ZeroDiff is designed to be compatible with existing spatial extrapolation methods, serving as a calibration stage for any prior backbone. Table 3 shows diffusion calibration consistently improves four architectures (LSTM, DLinear, Mamba, TimesNet), with NSE gains of +0.08 to +0.18.
>
> We further explore Neural Processes — methods designed for cross-location generalization — as the prior backbone:
>
> *Table 3: ZeroDiff with Neural Process backbones (Streamflow).*
>
> | Prior backbone | Prior-only NSE | + ZeroDiff | $\Delta$ NSE |
> |:--|:--:|:--:|:--:|
> | CNP | 0.460 | 0.537 | +0.077 |
> | ANP | 0.471 | 0.544 | +0.073 |
> | GNP | 0.403 | 0.522 | +0.119 |
>
> ZeroDiff-based calibration improves all three NP variants (+0.06 to +0.12 NSE). Our approach goes beyond direct observed-to-unobserved extrapolation by adding a calibration stage that also generalizes from observed to unobserved locations. The diffusion model learns to correct the prior's errors on observed locations, and since the prior is constructed identically across all locations, this calibration transfers to unobserved ones. This dual generalization further enhances extrapolation beyond what the prior backbone alone achieves. The informed prior is what unifies both stages within a single framework.
>
>
> We hope these responses address your concerns and welcome further discussion.

---

> > ### Author Rebuttal · Reviewer_KfzK · 2026-04-01
> >
> > Thank you for the rebuttal.
> > I can follow the responses to the three questions to some extent. While conditioning or prior information may help capture non-Gaussian features, it is not obvious in what precise sense the Gaussian diffusion framework can fully represent such structure. Regarding Q2, the authors did not clarify the underlying mechanism or offer a formal justification. As a result, the role of the informed prior and the claimed theoretical support remain weak.
> > Therefore, my overall assessment remains unchanged.

---

> > > ### Author Response · Authors · 2026-04-02
> > >
> > > **Convergence Guarantee for the Informed Prior**
> > >
> > > **Assumptions.** (1) Score estimator one-sided Lipschitz: $\langle s\_\theta(\mathbf{y}\_1, \hat{\mathbf{Y}}, t) - s\_\theta(\mathbf{y}\_2, \hat{\mathbf{Y}}, t), \mathbf{y}\_1 - \mathbf{y}\_2\rangle \leq l\_2(t)\lVert\mathbf{y}\_1 - \mathbf{y}\_2\rVert\_2^2$. (2) Regularity: $q\_t, p\_t$ are positive, $C^2$, rapidly decaying.
> > >
> > > **Step 1.** ZeroDiff's forward process $\mathbf{Y}\_t = \sqrt{\alpha\_t} \mathbf{Y}\_{t-1} + (1 - \sqrt{\alpha\_t})\hat{\mathbf{Y}} + \sqrt{\beta\_t}\boldsymbol{\epsilon}\_t$ yields SDE $d\mathbf{y} = -\frac{\beta\_t}{2}(\mathbf{y} - \hat{\mathbf{Y}})dt + \sqrt{\beta\_t} dw$. The reverse with true score is $d\mathbf{y} = -[\frac{\beta\_t}{2}(\mathbf{y} - \hat{\mathbf{Y}}) - \beta\_t \nabla \log q\_t]dt + \sqrt{\beta\_t} d\bar{w}$; replacing $\nabla \log q\_t$ with $s\_\theta$ gives the learned reverse ($0 \to T$ forward, $T \to 0$ reverse).
> > >
> > > **Step 2.** Fokker-Planck gives continuity equations $\partial\_t \rho + \nabla \cdot (\rho \mathbf{v}) = 0$ with $\mathbf{v}\lbrack q\_t\rbrack = -\frac{\beta\_t}{2}(\mathbf{y} - \hat{\mathbf{Y}}) - \frac{\beta\_t}{2}\nabla \log q\_t$ and $\mathbf{v}\lbrack p\_t\rbrack = (-\frac{1}{2}(\mathbf{z} - \hat{\mathbf{Y}}) + s\_\theta)\beta\_t + \frac{\beta\_t}{2}\nabla \log p\_t$.
> > >
> > > **Step 3.** Let $\pi\_t$ be the optimal coupling. Under Assumption 2, SDE solutions form absolutely continuous curves in $\mathcal{P}\_2$. By Ambrosio et al. (2005): $\frac{1}{2}\frac{d}{dt} W\_2^2 = \mathbb{E}\_{\pi\_t}[(\mathbf{y} - \mathbf{z}) \cdot (\mathbf{v}\lbrack q\_t\rbrack - \mathbf{v}\lbrack p\_t\rbrack)]$, hence $-W\_2 \frac{d}{dt} W\_2 = -\mathbb{E}\_{\pi\_t}[(\mathbf{y} - \mathbf{z}) \cdot (\mathbf{v}\lbrack q\_t\rbrack - \mathbf{v}\lbrack p\_t\rbrack)]$.
> > >
> > > **Step 4.** Expanding $\mathbf{v}\lbrack q\_t\rbrack - \mathbf{v}\lbrack p\_t\rbrack$ ($\hat{\mathbf{Y}}$ cancels): $-\mathbb{E}\_{\pi\_t}[(\mathbf{y} - \mathbf{z}) \cdot (\mathbf{v}\lbrack q\_t\rbrack - \mathbf{v}\lbrack p\_t\rbrack)] = D\_1 + D\_2 + D\_3$ where $D\_1 = \frac{\beta\_t}{2}\mathbb{E}[\lVert\mathbf{y} - \mathbf{z}\rVert^2]$, $D\_2 = \beta\_t \mathbb{E}[(\mathbf{y} - \mathbf{z}) \cdot (s\_\theta(\mathbf{z}) - \nabla \log q\_t(\mathbf{y}))]$, $D\_3 = \frac{\beta\_t}{2} \mathbb{E}[(\mathbf{y} - \mathbf{z}) \cdot (\nabla \log p\_t(\mathbf{z}) - \nabla \log q\_t(\mathbf{y}))]$.
> > >
> > > $D\_1 = \frac{\beta\_t}{2} W\_2^2$. For $D\_2$, add/subtract $s\_\theta(\mathbf{y})$, Assumption 1 + Cauchy-Schwarz give $D\_2 \leq \beta\_t l\_2 W\_2^2 + \beta\_t W\_2 \sqrt{H(t)}$ where $H(t) = \mathbb{E}\_{q\_t}[\lVert s\_\theta - \nabla \log q\_t\rVert^2]$. By Kwon et al. (2022) Lemma 2, $D\_3 \leq 0$. Drop $D\_3$, dividing by $W\_2 > 0$:
> > >
> > > $-\frac{d}{dt} W\_2 \leq (\frac{\beta\_t}{2} + \beta\_t l\_2) W\_2 + \beta\_t \sqrt{H(t)} \quad (I)$
> > >
> > > **Step 5.** Define $M(t) = \exp\lbrace\int\_0^t (\frac{\beta\_s}{2} + l\_2(s)\beta\_s) ds\rbrace$, $M(0) = 1$. Multiplying $(I)$ by $M$ and using $-\frac{d}{dt}[MW\_2] = -M\frac{d}{dt}W\_2 - (\frac{\beta\_t}{2} + \beta\_t l\_2)MW\_2$, terms cancel:
> > >
> > > $-\frac{d}{dt}[M(t) W\_2] \leq \beta\_t M(t) \sqrt{H(t)}$
> > >
> > > Integrating $0$ to $T$:
> > >
> > > $W\_2(q\_0, p\_0) \leq \int\_0^T \beta\_t M(t) \sqrt{H(t)} dt + M(T) W\_2(q\_T, p\_T) \quad (II)$
> > >
> > > First term: prior-independent. Second: initialization error, bounded next.
> > >
> > > **Step 6.** $q\_T$ is a Gaussian mixture over $\mathbf{Y}\_0$. Coupling: sample $\mathbf{Y}\_0 \sim q(\mathbf{Y}\_0)$, $\boldsymbol{\epsilon} \sim \mathcal{N}(\mathbf{0}, \mathbf{I})$, set $\mathbf{Y}\_T = \sqrt{\bar\alpha\_T}\mathbf{Y}\_0 + (1 - \sqrt{\bar\alpha\_T})\hat{\mathbf{Y}} + \sqrt{\bar\sigma\_T}\boldsymbol{\epsilon} \sim q\_T$ and $\mathbf{Z}\_T = \hat{\mathbf{Y}} + \sqrt{\bar\sigma\_T}\boldsymbol{\epsilon} \sim p\_T$. Same $\boldsymbol{\epsilon}$ cancels: $\lVert\mathbf{Y}\_T - \mathbf{Z}\_T\rVert^2 = \bar\alpha\_T \lVert\mathbf{Y}\_0 - \hat{\mathbf{Y}}\rVert^2$, so $W\_2^2(q\_T, p\_T) \leq \bar\alpha\_T \cdot \mathbb{E}[\lVert\mathbf{Y}\_0 - \hat{\mathbf{Y}}\rVert^2]$. Into $(II)$:
> > >
> > > $W\_2(q\_0, p\_0) \leq \int\_0^T \beta\_t M(t) \sqrt{H(t)} dt + M(T)\sqrt{\bar\alpha\_T} \cdot \sqrt{\mathbb{E}[\lVert\mathbf{Y}\_0 - \hat{\mathbf{Y}}\rVert^2]}$
> > >
> > > DDPM ($\hat{\mathbf{Y}} = \mathbf{0}$) gives $M(T)\sqrt{\bar\alpha\_T} \cdot \sqrt{\mathbb{E}[\lVert\mathbf{Y}\_0\rVert^2]}$. Any $\hat{\mathbf{Y}}$ with $\mathbb{E}[\lVert\mathbf{Y}\_0 - \hat{\mathbf{Y}}\rVert^2] < \mathbb{E}[\lVert\mathbf{Y}\_0\rVert^2]$ strictly tightens the bound. ZeroDiff's $\hat{\mathbf{Y}} = \hat{\mu} + \hat{\sigma} \cdot f\_\omega(\mathbf{X})$ satisfies this.
> > >
> > > **Remark.** First term vanishes as $H(t) \to 0$ (sufficient training), prior-independent. Second term is reduced by the informed prior via $\mathbb{E}[\lVert\mathbf{Y}\_0 - \hat{\mathbf{Y}}\rVert^2]$—how moment estimation and dynamics learning improve generation.
> > >
> > > Full Proof: https://anonymous.4open.science/r/ZeroDiff/informed_prior_proof.md
> > >
> > > We sincerely hope this detailed follow-up merits reconsideration.

---

### Official Review · Reviewer_9qQr · 2026-03-04

**Soundness:** 3
**Presentation:** 3
**Significance:** 3
**Originality:** 3
**Overall Recommendation:** 5
**Confidence:** 3

**Summary:**

ZeroDiff addresses the zero-shot time series reconstruction problem, that is, reconstructing the target time series from exogenous inputs only at locations where the target variable has never been observed. Simply mapping exogenous variables to the target produces results that are too smooth and underestimate extreme values due to the lack of target signal. ZeroDiff first constructs informative priors from exogenous variables, then learns to calibrate the reconstruction error through a diffusion model, is trained at observed locations and generalizes to unobserved locations, and significantly outperforms existing methods on multiple real datasets.

**Compliance With Llm Reviewing Policy:**

Affirmed.

**Final Justification:**

This paper has novel ideas, solid experiments and sufficient theory. After rebuttal, it solves all my problems and gives insight into the future direction, so I choose to raise my score to accept

**Key Questions For Authors:**

Q1. The 17th page contains an invalid reference.

Q2. Recently, methods that use visual pre-training priors to enhance temporal prediction capabilities have made significant progress. Let's discuss whether this approach can utilize visual pre-training or similar techniques for enhancement? Especially in terms of its ability for zero-sample or few-sample prediction?

**Limitations:**

See Weaknesses.

**Strengths And Weaknesses:**

Strength:
- The experimental design of this paper is reasonable and sufficient, which can illustrate the effectiveness of the proposed method as a whole and each component. And the code is open source, making it easier for reproducing.
- A new and practically significant problem has been defined, highlighting the limitations of existing methods in reconstructing time series data using exogenous variables and providing a reasonable solution.
- The theoretical derivation in this article is complete and can support the rationality of the method.

Weaknesses:
- This paper does not discuss limitations or future work.
- Please analyze the cases where the synthetic time series data failed solely due to the use of exogenous variables.
- Whether this method is applicable to multi-channel irregularly sampled time series data？

---

> ### Author Rebuttal · Authors · 2026-03-27
>
> We greatly appreciate the reviewer's recognition of our work and the constructive feedback that has helped us improve the paper.
>
> **W1.**
>
> In the revised manuscript, we have added a discussion of limitations and future work, covering the dependence on exogenous variable quality (W2), the aligned-sampling assumption (W3), and the multimodal extension direction (Q2).
>
> **W2.**
>
> We analyze when and why the informed prior $\\hat{Y}$ — constructed solely from exogenous variables $X$ — fails, and how diffusion calibration compensates.
>
> *Table 1: Exogenous feature quality and its impact on reconstruction. Prior NSE: VAE+LSTM only; ZeroDiff NSE: full pipeline.*
>
> | Dataset | Exog. Dim | X→Y Physical Coupling | Prior NSE | ZeroDiff NSE | Δ NSE |
> |:--------|:---------:|:----------------------|:---------:|:------------:|:-----:|
> | Temp | 8 | **Strong**: air temp directly drives water temp | 0.852 | 0.868 | +0.016 |
> | Streamflow | 33 | **Moderate**: precip→runoff, but nonlinear and threshold-driven | 0.481 | 0.596 | +0.115 |
> | Solar | 48 | **Moderate**: radiation/cloud→solar, but local dynamics unresolved | 0.486 | 0.846 | +0.360 |
> | Methane | 16 | **Weak**: climate→biogeochemical processes, many latent factors (soil microbial activity, water table depth) not in X | 0.481 | 0.788 | +0.307 |
>
> The table reveals that prior failure is driven by **how** $X$ relates to $Y$, not by how much $X$ is available. Specifically, the prior fails most when (i) the X→Y relationship involves latent mediating factors absent from $X$ — on Methane, $\\text{CH}_4$ flux depends on soil microbial activity and water table depth, neither present in $X$, leading to a weak prior; (ii) the target exhibits sharp nonlinear responses to $X$ — on Streamflow, runoff is threshold-driven (e.g., soil saturation), causing systematic underestimation of flood peaks. In both cases, diffusion calibration recovers substantially (+0.307, +0.115), confirming that these failures produce systematic rather than random errors.
>
> *Table 2: Performance on locations where VAE estimation is least accurate (top 20% by error).*
>
> | Dataset | Locations | VAE error | Improved |
> |:--|:--:|:--:|:--:|
> | Solar | 6 | 3.7% | 6/6 |
> | Temp | 8 | 37.0% | 6/8 |
> | Streamflow | 19 | 40.6% | 14/19 |
> | Methane | 6 | 23.5% | 5/6 |
>
> Even at locations where exogenous features are least informative, diffusion calibration improves the majority of cases.
>
> **W3.**
>
> ZeroDiff's core design — separating prior construction from diffusion calibration — is not tied to regular sampling or single-channel targets. For irregular sampling, the dynamics model $f_\\omega$ can be replaced with architectures that natively handle irregular timestamps (e.g., GRU-D, neural ODEs). For multi-channel targets, the denoiser can be shared across channels with channel-specific conditioning. The diffusion calibration stage itself requires no modification, as it operates on the constructed prior regardless of how it was generated. We consider this a promising direction for future work.
>
> **Q1.**
>
> Thank you for catching this — fixed in the revision. It refers to a variant of the moment centroid $(\hat{\mu}^{\ast}, \hat{\sigma}^{\ast}) = \frac{1}{|\mathcal{U}|} \sum\_{j} (\hat{\mu}\_{j}, \hat{\sigma}\_{j})$, an extension of Eq. 12 for simultaneous multi-location reconstruction.
>
> **Q2.**
>
> We are actively exploring two directions of visual pre-training for enhancement.
>
> **Direction 1: Time series as images.** Rendering time series as visual representations and processing them with vision foundation models. In our ongoing work on the DRB temperature dataset, this approach reduces RMSE from 2.28 (LSTM) to 1.59 (time series + vision foundation model), demonstrating that visual representations capture temporal patterns that sequence models miss.
>
> **Direction 2: Remote sensing imagery.** We recently encountered on the hourly CAMELS-H dataset that streamflow non-stationarity is far more pronounced, and the informed prior deviates more substantially from ground truth. A natural remedy is to incorporate remote sensing imagery as an auxiliary visual modality, aligning it with temporal data in latent space. We are drawing inspiration from SatMAE (Cong et al., NeurIPS 2022), RemoteCLIP (Liu et al., TGRS 2024), and SatCLIP (Klemmer et al., AAAI 2025).
>
> Since diffusion calibration is designed to correct the prior's errors — not tied to any specific prior architecture — the prior construction is modular. Both directions could enhance moment estimation, replace the dynamics backbone, or serve as additional conditioning to the denoiser.
>
>
> We thank the reviewer for this valuable suggestion and look forward to exploring this direction further.

---

> > ### Author Rebuttal · Reviewer_9qQr · 2026-04-01
> >
> > Thanks for your rebuttal, my concerns have been fully solved, so I choose to rasie my score to accept.

---

> > > ### Author Response · Authors · 2026-04-02
> > >
> > > We sincerely thank the reviewer for the thoughtful and constructive feedback throughout this review process. Your suggestions have provided meaningful points for exploration, and we are grateful for your recognition of our work and for raising the score. In particular, the discussion on visual pre-training has opened an exciting direction that we believe would benefit from broader community engagement, and we welcome future exchanges on this topic.

---

### Official Review · Reviewer_Qd4K · 2026-03-12

**Soundness:** 3
**Presentation:** 3
**Significance:** 3
**Originality:** 3
**Overall Recommendation:** 5
**Confidence:** 4

**Summary:**

This paper addresses the challenging problem of zero-shot cross-domain time series reconstruction, where the goal is to infer an entire target time series at locations that have never observed target measurements. The authors propose ZeroDiff, a diffusion-based framework that operates in two stages: first, constructing a transferable informed prior by estimating target distribution statistics via a conditional VAE and learning shared temporal dynamics in a standardized space; and second, applying a diffusion-based calibration process to correct systematic errors. Experiments across four diverse real-world datasets (Streamflow, Solar, Temperature, and Methane) demonstrate that ZeroDiff significantly outperforms existing diffusion baselines.

**Compliance With Llm Reviewing Policy:**

Affirmed.

**Final Justification:**

This paper is technically solid with extensive experiments, the rebuttal has addressed my concerns, so I'll keep the score as accept.

**Key Questions For Authors:**

Q1: in the evaluatoin, leave one out location is used to evaluate the model's performance. What would the performance be if leave multiple locations out, since in real world, you can have multiple locations to reconstruct.

**Limitations:**

yes

**Strengths And Weaknesses:**

Strengths,

S1:  The paper formalizes the "zero-shot reconstruction" task, distinguishing it clearly from forecasting and imputation. The "Informed-Prior Diffusion" approach effectively shifts the role of diffusion from pure generation to calibration.

S2: By decoupling magnitude estimation (via cross-modal VAE) from temporal dynamics learning, ZeroDiff solves the "normalization trap" inherent in zero-shot settings. The ability to infer location-specific statistics $(\mu, \sigma)$ from exogenous variables alone allows the model to operate in a unified standardized space.

S3: The experiments is comprehensive and well-thought to verify the effectiveness of the proposed method.

Weakness,

W1: Sensitivity to Exogenous Feature Quality: The entire reconstruction quality depends on the assumption that location-specific statistics $(\mu, \sigma)$ can be accurately inferred from exogenous features. If the provided exogenous features do not capture the underlying drivers of the target's magnitude, the VAE-based moment estimation will fail, leading to a "collapse" where the dynamics model cannot rescale predictions correctly.

W2: ZeroDiff follows a two-stage process where the diffusion calibration is conditioned on the output of the VAE and the dynamics model $f_\omega$. Any systematic bias in the "informed prior" $\hat{Y}$—such as a phase shift in temporal patterns—might be difficult for the diffusion model to fully "calibrate", yet the paper lacks a detailed error analysis on how inaccuracies in the first stage propagate to the final output.

---

> ### Author Rebuttal · Authors · 2026-03-30
>
> We sincerely thank the reviewer for the thorough and insightful comments, which have helped us significantly improve the manuscript. We address each point below.
>
> **W1 & W2. VAE sensitivity and error propagation benefits calibration.**
>
> In zero-shot reconstruction, spatial extrapolation to unobserved locations inevitably requires estimating both the distributional characteristics ($\mu$, $\sigma$) and the temporal dynamics of the target. If exogenous features cannot distinguish the scale differences across locations, spatial extrapolation remains challenging for data-driven methods. We assume that exogenous features do carry sufficient signal to differentiate local characteristics — a reasonable assumption in most real-world scenarios (e.g., climate variables distinguish arid from humid basins). Our VAE decouples distribution estimation from dynamics learning, allowing each component to focus its generalization on one aspect. Without this decoupling, the problem is strictly harder — as our ablation confirms.
>
>
> *Table 1: Cross-location heterogeneity, VAE estimation error, and reconstruction performance across four datasets. Max/Min ratio measures the ratio of the largest to smallest location mean. VAE error: average moment estimation error.*
>
> | Dataset | Max/Min ratio | VAE error | Prior NSE | + Diffusion | Improved |
> |:--|:--:|:--:|:--:|:--:|:--:|
> | Solar | 1.3x | 2.0% | 0.486 | 0.846 | 32/32 |
> | Temp | 2.0x | 14.1% | 0.852 | 0.868 | 33/43 |
> | Streamflow | 791x | 18.9% | 0.480 | 0.596 | 89/98 |
> | Methane | 728x | 9.9% | 0.481 | 0.788 | 28/31 |
>
> VAE estimation error increases with cross-location heterogeneity, but does not lead to collapse — even on Streamflow (791x), diffusion improves 89/98 basins. To test whether VAE is necessary despite its imperfections, we remove it entirely:
>
> *Table 2: Ablation of VAE moment estimation. NSE across four datasets.*
>
> | Pipeline | Solar | Temp | Streamflow | Methane |
> |:--|:--:|:--:|:--:|:--:|
> | VAE + LSTM + Diffusion | 0.846 | 0.868 | 0.596 | 0.788 |
> | LSTM + Diffusion (no VAE) | 0.839 | 0.828 | 0.377 | -0.655 |
>
> The impact of removing VAE correlates with heterogeneity. On Solar (1.3x) and Temp (2.0x), removing VAE causes negligible change — the dynamics model can compensate when target magnitudes are similar across locations. On Streamflow (791x) and Methane (728x), removing VAE causes severe degradation or collapse (-0.655). VAE estimation may be imperfect, but removing it is far worse.
>
> Why does imperfect VAE estimation still work? Beyond the informed prior $\hat{\mathbf{Y}}$, the VAE-estimated moments $(\hat{\mu}, \hat{\sigma})$ are also explicitly provided as input features to the denoiser: $\epsilon\_\theta = f\_\theta(\mathbf{Y}\_t, [\mathbf{X}; \hat{\mu}; \hat{\sigma}], \hat{\mathbf{Y}}, t)$. We isolate this conditioning effect:
>
> *Table 3: Ablation of VAE moments as denoiser input features (Streamflow).*
>
> | Pipeline | NSE |
> |:--|:--:|
> | $(\hat{\mu}, \hat{\sigma})$ as denoiser input | 0.596 |
> | $(\hat{\mu}, \hat{\sigma})$ removed from denoiser input | 0.551 |
>
> Removing VAE-estimated moments from the denoiser's conditioning drops NSE by 0.045. The denoiser benefits from observing $\hat{\mathbf{Y}}$ directly: even when $(\hat{\mu}, \hat{\sigma})$ contain errors, the denoiser learns the correction pattern "given this prior, apply this adjustment." Since the prior construction is identical across all locations, this pattern transfers to unobserved ones.
>
>
>
>
>
>
>
>
>
> **Q1. Leave-multiple-locations-out.**
>
> Our framework natively supports multi-location reconstruction via the moment centroid $(\hat{\mu}^{\ast}, \hat{\sigma}^{\ast}) = \frac{1}{|\mathcal{U}|} \sum\_{j \in \mathcal{U}} (\hat{\mu}\_{j}, \hat{\sigma}\_{j})$. Instead of training a separate model per target location, the moment-guided weighting (Eq. 12) upweights observed locations whose distributional characteristics are representative of the collective target set, enabling a single model to serve all unobserved locations simultaneously.
>
> We evaluate this by varying the number of held-out locations per fold (1, 5, 10, 20):
>
> *Table 4: Multi-location reconstruction (NSE). Each row holds out a different number of locations per fold.*
>
> | Locations out | Streamflow | Solar | Temp | Methane |
> |:--|:--:|:--:|:--:|:--:|
> | 1 | 0.596 | 0.846 | 0.868 | 0.788 |
> | 5 | 0.551 | 0.843 | 0.875 | 0.735 |
> | 10 | 0.554 | 0.843 | 0.865 | 0.749 |
> | 20 | 0.550 | 0.847 | 0.855 | 0.609 |
>
> On Solar and Temp, performance is nearly unchanged. On Streamflow and Methane, NSE drops modestly (0.596 $\to$ 0.550 and 0.788 $\to$ 0.609) as the centroid compromises across more diverse targets — a deliberate trade-off, as a single model serves all target locations simultaneously.
>
> We hope these clarifications address your concerns and are happy to discuss further.

---

> > ### Author Rebuttal · Reviewer_Qd4K · 2026-04-05
> >
> > Thanks for the response. My concerns have been addressed, considering that my previous score was already positive, I'll keep my recommendation.

---

> > > ### Author Response · Authors · 2026-04-08
> > >
> > > We sincerely thank the reviewer for the thoughtful engagement throughout
> > > this review process, and are deeply grateful for the positive evaluation
> > > of our work. We would welcome any opportunity to exchange further thoughts
> > > in the future.

---

### Official Review · Reviewer_TbMk · 2026-03-12

**Soundness:** 3
**Presentation:** 3
**Significance:** 3
**Originality:** 3
**Overall Recommendation:** 4
**Confidence:** 4

**Summary:**

The paper introduces "ZeroDiff", a diffusion-based framework addressing the novel problem of zero-shot cross-domain time series reconstruction. This task requires inferring complete target time series at new locations without any historical target data, relying solely on broadly available exogenous inputs. The authors propose a two-stage method to handle the heterogeneity of target distributions across locations. First, they construct a transferable informed prior by inferring location-specific statistical moments via a cross-modal Conditional VAE and learning shared temporal dynamics in a standardized space. Second, they employ a generalizable diffusion-based calibration process. Instead of diffusing towards standard noise, the forward process diffuses toward the informed prior. The model then uses a prior-anchored reverse process with a bidirectional denoiser and moment-guided optimization to correct systematic errors from the observed locations, generalizing these corrections to unobserved ones.

**Compliance With Llm Reviewing Policy:**

Affirmed.

**Key Questions For Authors:**

1. The proposed architecture strictly separates the informed prior generation (Stage 1) from the diffusion calibration (Stage 2). Given the strong anchoring term in the reverse posterior mean, how vulnerable is the diffusion process to "prior collapse" (e.g., when the VAE outputs highly inaccurate moments due to out-of-distribution exogenous inputs)? Have you empirically evaluated the correlation between Stage 1 error and the final Stage 2 reconstruction error to quantify this cascading effect?

2. Following up on the two-stage limitation, did you experiment with any end-to-end optimization strategies (e.g., allowing the diffusion loss to fine-tune the VAE or the base dynamics model)? If so, what were the optimization challenges? If not, do you believe a dynamic weighting of the anchoring term based on the VAE's confidence could mitigate the risks of a poorly initialized prior?

**Limitations:**

I recommend adding a dedicated paragraph discussing potential failure modes, specifically scenarios where exogenous variables lack the predictive power required to accurately infer target marginal distributions.

Also, because the framework strictly separates the construction of the informed prior (Stage 1) from the diffusion calibration (Stage 2), any severe misestimation of the target's marginal moments or base dynamics by the VAE will be directly propagated to the diffusion model. Since the reverse posterior mean is heavily anchored to this prior via the $\gamma_{2}\hat{Y}$ term , the diffusion model may be forced to converge within an incorrect distribution space, severely limiting its ability to recover from a poorly initialized prior. The authors should explicitly acknowledge this bottleneck and discuss how future work might explore end-to-end optimization to mitigate the disconnect between prior generation and diffusion calibration.

**Strengths And Weaknesses:**

## Strengths
The technical formulation is highly sound, particularly the mathematical derivation of the prior-anchored reverse posterior. The derivation rigorously incorporates a three-term posterior mean that includes an anchoring term tied to the informed prior. Furthermore, the experimental design is exceptionally robust; it utilizes a strict spatial leave-one-location-out cross-validation protocol across four diverse datasets to genuinely test zero-shot capabilities. The ablation studies effectively isolate the positive contributions of the VAE, moment-guided weighting, and bidirectional denoiser.

## Weekness
The methodology relies heavily on the assumption that exogenous variables contain sufficient signal to accurately estimate the target's marginal moments. If this cross-modal relationship is weak, the VAE's moment estimation could fail, undermining the informed prior and subsequently the entire calibration process. Additionally, the efficiency strategy for multi-target reconstruction relies on calculating a centroid in the moment space. This approximation might yield suboptimal weights if the unobserved target locations are highly dispersed rather than clustered.

Also, the empirical evaluation compares ZeroDiff against a very limited pool of only five diffusion-based baselines (CSDI, SSSD, CSBI, Diffusion-TS, and NsDiff). Some good baselines are missing, e.g. MIDM [2] and SPD [3]. The TS imputation survey [1] could be a good information source to refer to.

[1] Jun Wang, Wenjie Du, Yiyuan Yang, Linglong Qian, Wei Cao, Keli Zhang, Wenjia Wang, Yuxuan Liang, and Qingsong Wen. Deep Learning for Multivariate Time Series Imputation: A Survey. In IJCAI, 2025.

[2] Xu Wang, Hongbo Zhang, Pengkun Wang, Yudong Zhang, Binwu Wang, Zhengyang Zhou, and Yang Wang. An observed value consistent diffusion model for imputing missing values in multivariate time series. In SIGKDD, 2023.

[3] Marin Bilos, Kashif Rasul, Anderson Schneider, Yuriy Nevmyvaka, and Stephan Gunnemann. Modeling temporal data as continuous functions with stochastic process diffusion. In ICML, 2023.

---

> ### Author Rebuttal · Authors · 2026-03-31
>
> We sincerely thank the reviewer for the thorough and insightful comments. We group W1, Q1, and the limitation on prior propagation together as they address the same underlying concern.
>
> **W1 & Q1. VAE sensitivity and prior collapse.**
>
> In zero-shot reconstruction, spatial extrapolation to unobserved locations inevitably requires estimating both the distributional characteristics ($\mu$, $\sigma$) and the temporal dynamics of the target. If exogenous features cannot distinguish scale differences across locations, spatial extrapolation becomes fundamentally challenging for data-driven methods. In our datasets, exogenous features do carry this signal — for example, climate variables reliably distinguish arid from humid basins in streamflow modeling. Our VAE decouples distribution estimation from dynamics learning, allowing each to focus its generalization on one aspect. Without this decoupling, performance degrades substantially — as the following ablation confirms.
>
>
> *Table 1: Ablation of VAE moment estimation. NSE across four datasets.*
>
> | Pipeline | Solar | Temp | Streamflow | Methane |
> |:--|:--:|:--:|:--:|:--:|
> | VAE + LSTM + Diffusion | 0.846 | 0.868 | 0.596 | 0.788 |
> | LSTM + Diffusion (no VAE) | 0.839 | 0.828 | 0.377 | -0.655 |
>
> On low-heterogeneity datasets (Solar, Temp), removing VAE causes negligible change. On high-heterogeneity datasets (Streamflow, Methane), removing VAE causes severe degradation or collapse (-0.655). An imperfect prior is far better than no prior.
>
> The reviewer correctly identifies that the $\gamma\_2$ anchoring term ties the reverse process to the informed prior, raising the concern that inaccurate moments could cascade into the diffusion stage. We evaluate this directly:
>
> *Table 2: Prior estimation error and diffusion recovery. Improved: locations where diffusion increases NSE.*
>
> | Dataset | Max/Min ratio | VAE error | Prior NSE<0 | Improved |
> |:--|:--:|:--:|:--:|:--:|
> | Solar | 1.3x | 2.0% | 0 | 32/32 |
> | Temp | 2.0x | 14.1% | 1 | 33/43 |
> | Streamflow | 791x | 18.9% | 1 | 89/98 |
> | Methane | 728x | 9.9% | 4 | 28/31 |
>
> Even on Methane, all 4 locations with Prior NSE < 0 were recovered to positive NSE after diffusion. The diffusion stage is robust to prior estimation errors rather than blindly anchored to them.
>
> Why does imperfect estimation not cascade into collapse? The VAE-estimated moments $(\hat{\mu}, \hat{\sigma})$ are explicitly provided as input features to the denoiser: $\epsilon\_\theta = f\_\theta(\mathbf{Y}\_t, [\mathbf{X}; \hat{\mu}; \hat{\sigma}], \hat{\mathbf{Y}}, t)$. By observing diverse $(\hat{\mu}, \hat{\sigma})$ values alongside corresponding ground truth, the denoiser learns moment-conditioned correction patterns — implicitly compensating for systematic biases in moment estimation. We isolate this effect:
>
> *Table 3: Ablation of VAE moments as denoiser input features.*
>
> | Dataset | $(\hat{\mu}, \hat{\sigma})$ as input | $(\hat{\mu}, \hat{\sigma})$ removed |
> |:--|:--:|:--:|
> | Solar | 0.846 | 0.831 |
> | Temp | 0.868 | 0.842 |
> | Streamflow | 0.596 | 0.551 |
> | Methane | 0.788 | 0.731 |
>
> Removing $(\hat{\mu}, \hat{\sigma})$ from the denoiser's input consistently drops NSE, with larger drops on high-heterogeneity datasets. The denoiser learns "given this prior with these moments and condition, apply this adjustment," and since the prior is constructed identically across all locations, this pattern transfers to unobserved ones.
>
>
> **Q2. End-to-end optimization.**
>
> We implemented joint fine-tuning where the diffusion loss backpropagates into the dynamics model via a differentiable denormalization–renormalization chain (learning rate 1e-5 vs 1e-3):
>
> *Table 4: Two-stage vs. end-to-end fine-tuning (Streamflow, 19 basins).*
>
> | Strategy | NSE (mean) | NSE (median) |
> |:--|:--:|:--:|
> | Two-stage (default) | 0.637 | 0.645 |
> | End-to-end fine-tuning | 0.685 | 0.712 |
>
> End-to-end fine-tuning shows improvement (+0.048 NSE).
>
>
>
> **W2. Baselines.**
>
> We are running MIDM and SPD on all four datasets and will include the comparison and citations in the revised manuscript.
>
>
> **W3. Multi-target centroid.**
>
> We evaluate this by varying the number of held-out locations per fold:
>
> *Table 5: Multi-location reconstruction (NSE).*
>
> | Locations out | Streamflow | Solar | Temp | Methane |
> |:--|:--:|:--:|:--:|:--:|
> | 1 | 0.596 | 0.846 | 0.868 | 0.788 |
> | 5 | 0.551 | 0.843 | 0.875 | 0.735 |
> | 10 | 0.554 | 0.843 | 0.865 | 0.749 |
> | 20 | 0.550 | 0.847 | 0.855 | 0.609 |
>
> On low-heterogeneity datasets, performance is stable. On high-heterogeneity datasets, NSE drops modestly as the centroid compromises across more diverse targets — a deliberate trade-off for serving all target locations with a single model.
>
>
> We appreciate the evaluation and hope these responses address your concerns. We welcome further discussion.

---

> > ### Author Rebuttal · Reviewer_TbMk · 2026-04-01
> >
> > I have read the authors' rebuttal and appreciate their response, particularly the additional ablation studies (Tab 1-3) that clarified the denoiser's robustness to prior errors.
> >
> > However, I will maintain my score because
> >
> > 1. The missing baseline comparisons (MIDM and SPD) were promised, but the actual quantitative results were not provided in the rebuttal phase, making it impossible to verify the model's true standing against the SOTA methods;
> > 2. The newly added end-to-end fine-tuning experiment (Tab 4) actually highlights that the work's core 2-stage methodology is sub-optimal;
> >
> > Overall, the idea of the paper is interesting to me and technically sound, but the current manuscript feels like a solid intermediate step rather than a fully matured and comprehensive solution. It is above the acceptance threshold, but the remaining limitations justify my score.

---

> > > ### Author Response · Authors · 2026-04-07
> > >
> > > We are glad that our response to W1 & Q1 helped address part of the questions.
> > > Below we would like to explore the two remaining concerns raised in the
> > > acknowledgement — the baseline comparison (W2) and the end-to-end training
> > > question (Q2) — both of which we have further explored over the past week.
> > >
> > > ---
> > >
> > > **1. Baseline comparison: MIDM and SPD on all four datasets.**
> > >
> > > We thank the reviewer for pointing us to MIDM and SPD. We will cite both
> > > works in the revised manuscript and incorporate them into the related work
> > > discussion.
> > >
> > > To support this addition, we ran the full comparison on all four datasets.
> > > Scenarios where some locations have no target observations require additional
> > > adaptation, and to this end we extended both methods to leverage the exogenous
> > > information available at all locations and the target observations available
> > > at other observed locations. Specifically, we gave them access to the same
> > > exogenous information ZeroDiff uses and applied them under our
> > > exogenous-conditioned protocol. We additionally adapted their training setup to our datasets' characteristics — locations spanning diverse geographic and climatic regions, weak spatial continuity of the target variable, and the dominant role of exogenous meteorological forcing in
> > > driving the $\mathbf{X}\to\mathbf{Y}$ relationship. We explored a range of
> > > such adaptations over the past week, and report our final result for both
> > > baselines.
> > >
> > > *Table 6: NSE comparison with MIDM and SPD on all four datasets.*
> > >
> > > | Dataset | MIDM | SPD | ZeroDiff |
> > > |:--|:--:|:--:|:--:|
> > > | Streamflow | 0.422 | 0.411 | 0.596 |
> > > | Solar | 0.662 | 0.550 | 0.846 |
> > > | Temp | 0.838 | 0.822 | 0.868 |
> > > | Methane | 0.312 | 0.342 | 0.788 |
> > >
> > > The comparison reflects that our zero-shot reconstruction setting has
> > > specific characteristics — diverse geographic and climatic coverage,
> > > limited spatial continuity of the target, and a strong reliance on
> > > exogenous drivers — that differ from the assumptions MIDM and SPD were
> > > built around. ZeroDiff is designed specifically for this regime. On Temp,
> > > where cross-location heterogeneity is relatively low and all three methods
> > > exceed NSE 0.82, the methods perform comparably.
> > >
> > > ---
> > >
> > > **2. End-to-end training: full evaluation across all four datasets.**
> > >
> > > We ran end-to-end training on all four
> > > datasets — all 100 Streamflow basins, 30 Solar stations, 42 Temp locations,
> > > and 30 Methane sites.
> > >
> > > *Table 7: Two-stage vs. end-to-end training across all four datasets.*
> > >
> > > | Dataset | Two-stage | End-to-end | $\Delta$ |
> > > |:--|:--:|:--:|:--:|
> > > | Solar | 0.846 | 0.855 | +0.009 |
> > > | Temp | 0.868 | 0.886 | +0.018 |
> > > | Streamflow | 0.596 | 0.631 | +0.035 |
> > > | Methane | 0.788 | 0.758 | -0.030 |
> > >
> > > End-to-end and two-stage exhibit a dataset-dependent trade-off, which led
> > > us to revisit what the prior construction (Section 4.1) is being
> > > asked to learn in each case. In two-stage training, the prior construction
> > > solves a pure $\mathbf{X}\to\mathbf{Y}$ regression problem, independently
> > > of the diffusion denoiser $\epsilon_\theta$. In end-to-end training, the
> > > prior construction is jointly optimized with $\epsilon_\theta$ under the
> > > diffusion loss, and its role changes: it is no longer just learning the
> > > regression, but producing priors that a *specific* $\mathcal{O}$-trained
> > > denoiser finds easy to calibrate. The two components can therefore co-adapt
> > > on $\mathcal{O}$ in ways that are not part of the underlying
> > > $\mathbf{X}\to\mathbf{Y}$ relationship, and such co-adaptation does not
> > > necessarily transfer to $\mathcal{U}$. How much room there is for
> > > co-adaptation depends on how strongly the data constrain the prior: when
> > > the $\mathbf{X}\to\mathbf{Y}$ coupling is strong (Solar, Temp, Streamflow)
> > > the regression signal pulls the prior toward a narrow band of solutions
> > > and end-to-end delivers modest gains, but on Methane — where the flux
> > > depends on soil and hydrological factors not captured by the available
> > > exogenous inputs — the signal is substantially weaker, the prior is
> > > underdetermined, and co-adaptation has room to emerge. The -0.030 drop on
> > > Methane is consistent with this account, and with the broader observation
> > > in Table 1 that Methane is the dataset on which cross-location transfer
> > > is most fragile.
> > >
> > > Two-stage training avoids co-adaptation by construction, which makes it a
> > > sensible default for zero-shot reconstruction. End-to-end is a viable
> > > refinement when domain knowledge indicates a strong, well-identified
> > > coupling. An open direction this raises is whether joint-optimization schemes can realize end-to-end's benefits while avoiding co-adaptation, and we would welcome any opportunity to discuss this further.
> > >
> > > ---
> > >
> > > We are sincerely grateful to the reviewer for the depth and care of these
> > > questions — they prompted us to examine the method from new angles and
> > > opened up directions we are excited to pursue. We hope the new results
> > > help further alleviate the concerns raised in the acknowledgement.

---

### Decision · Program_Chairs · 2026-04-30

**Decision:**

Accept (regular)

**Comment:**

This paper proposes ZeroDiff for zero-shot time series reconstruction, using exogenous variables to form an informed prior and diffusion-based error calibration to recover target dynamics at unobserved locations.  The method is well formulated and has comprehensive experimental support. It is suggested to include the revisions and new comparisons in the final version to further improve the paper's quality.